# SE3Set: Harnessing Equivariant Hypergraph Neural Networks for Molecular Representation Learning

## Abstract

In this paper, we develop SE3Set, an SE(3) equivariant hypergraph neural network architecture tailored for advanced molecular representation learning. Hypergraphs are not merely an extension of traditional graphs; they are pivotal for modeling high-order relationships, a capability that conventional equivariant graph-based methods lack due to their inherent limitations in representing intricate many-body interactions. To achieve this, we first construct hypergraphs via proposing a new fragmentation method that considers both chemical and three-dimensional spatial information of the molecular system. We then design SE3Set, which incorporates equivariance into the hypergraph neural network. This ensures that the learned molecular representations are invariant to spatial transformations, thereby providing robustness essential for accurate prediction of molecular properties. SE3Set has shown performance on par with state-of-the-art (SOTA) models for small molecule datasets like QM9 and MD17. It excels on the MD22 dataset, achieving a notable improvement of approximately 20% in accuracy across all molecules, which highlights the prevalence of complex many-body interactions in larger molecules. This exceptional performance of SE3Set across diverse molecular structures underscores its transformative potential in computational chemistry, offering a route to more accurate and physically nuanced modeling.

## 1 Introduction

Molecular representation (Mathews & Chaffee, 2012; David et al., 2020; Wigh et al., 2022) is pivotal for cheminformatics (Fourches et al., 2010), impacting the prediction of molecular properties in drug discovery and material science. Traditional descriptors like fingerprints capture basic structural and energetic aspects of molecules by considering mainly one- and two-body interactions. However, they often miss complex electronic correlations and collective behaviors important for understanding phenomena such as chemical reactivity and protein folding. To address this, advanced methods that include many-body interactions are crucial for a more comprehensive molecular characterization. These methods enhance the predictive capabilities of computational models by more accurately reflecting the intricate dynamics and properties of molecules, which are essential for a deeper understanding of their functionality and reactivity in cheminformatics.

Graph neural networks (GNNs) (Zhou et al., 2020; Wu et al., 2020) are a foundational tool for representing structured data in molecular sciences with atoms as nodes and chemical bonds as edges, respectively. GNN models excel in tasks ranging from property prediction to reaction simulation (Do et al., 2019; Xiong et al., 2021; Reiser et al., 2022). GNNs can capture higher-order molecular interactions through message passing (Gilmer et al., 2017) but face overfitting and inefficiency challenges (Godwin et al., 2021; Rusch et al., 2023). Architectural improvements in GNNs facilitate the modeling of complex interactions, overcoming some limitations of deep networks (Gasteiger et al., 2019; Schütt et al., 2021; Batzner et al., 2022). Advances demonstrate the potential of architectural enhancements in GNNs to represent complex interactions (Gasteiger et al., 2020; 2021; Thölke & De Fabritiis, 2021; Batatia et al., 2022; Musaelian et al., 2023; Wang et al., 2024), but efficiently integrating many-body interactions into these networks is an ongoing challenge (Wang et al., 2023).

To address the complexities of many-body interactions in molecular systems, hypergraphs offer a compelling alternative to complex GNN architectures. Hypergraphs, with hyperedges connecting multiple vertices, can naturally represent many-body phenomena like electronic delocalization and collective vibrations. This allows for a more accurate modeling of molecular intricacies beyond the limitations of traditional graphs. Integrating hypergraphs with machine learning, particularly through Hypergraph Neural Networks (HGNNs), is an emerging research area. HGNNs manage the flow of information across hyperedges, capturing complex multi-atom interactions and enriching molecular representations. This technique promises to balance model expressiveness with computational efficiency. By innately encoding many-body interactions, HGNNs stand to significantly advance cheminformatics, offering a new approach to molecular property prediction and simulation that resonates with the actual behavior of chemical systems.

In this work, we introduce SE3Set, an innovative approach that enhances traditional GNNs by exploiting hypergraphs for modeling many-body interactions, while ensuring SE(3) equivariant representations that remain consistent regardless of molecular orientation. Our key contributions are:

- A new fragmentation method for hypergraph construction that seamlessly integrates 2D chemical and 3D spatial information, enriching the molecular structure representation.

- The deployment of hypergraph neural networks to capture many-body interactions, providing a deeper insight into molecular behavior that surpasses conventional pairwise modeling.

- The incorporation of SE(3) equivariance within our hypergraph framework, guaranteeing orientation-independent molecular representations.

- SE3Set underwent a comprehensive benchmarking process, exhibiting comparable outcomes to state-of-the-art (SOTA) models on small molecule datasets QM9 and MD17. It demonstrated exceptional performance on the larger molecule dataset MD22, where higher-order interactions are more evident, surpassing SOTA models with a significant reduction in mean absolute errors (MAEs) by an average of roughly 20%. This confirms SE3Set's efficacy in capturing the complexity of molecular representations.

These advances establish SE3Set as a formidable tool for molecular representation learning, with implications for computational chemistry and beyond.

## 2 Related works

### 2.1 Graph neural networks

Message passing neural networks (MPNNs), a class of graph neural networks, are essential for learning node features by transmitting information along graph edges, a process crucial for interpreting structured data like molecules (Gilmer et al., 2017). Equivariant GNNs are especially important for molecular modeling. They adopt either group representation methods, aligning architectures to symmetry groups for improved interaction modeling (Thomas et al., 2018; Anderson et al., 2019; Fuchs et al., 2020; Batzner et al., 2022; Liao & Smidt, 2022; Liao et al., 2023; Musaelian et al., 2023), or direction-based methods that incorporate spatial information for accurate molecular representations (Schütt et al., 2017; Kindermans & Müller, 2018; Coors et al., 2018; Gasteiger et al., 2019; 2020; Schütt et al., 2021; Thölke & De Fabritiis, 2021; Gasteiger et al., 2021; Wang et al., 2022; Du et al., 2024; Aykent & Xia, 2024; Wang et al., 2024) and have been engineered to handle intricate up to five-body interactions (Wang et al., 2023).

### 2.2 Hypergraph neural networks

Hypergraph Neural Networks (HGNNs) enhance GNNs by incorporating multi-node hyperedges, better capturing complexity in data from various domains. They advance GNNs' implicit many-body interactions with methods like clique expansion for compatibility with existing algorithms (Agarwal et al., 2005; Zhou et al., 2006) and employ tensor techniques for improved hypergraph-based feature learning (Li et al., 2013; Pearson

& Zhang, 2014; Benson et al., 2017; Chien et al., 2021a; Tudisco et al., 2021). While equivariant HGNNs adeptly handle node permutations, preserving data symmetries (Kim et al., 2021; 2022a), they often miss 3D spatial transformations, crucial for physical system modeling. In computational chemistry, hypergraph algorithms simulate complex behaviors and optimize molecules through hypergraph grammar (Cui et al., 2023; Tavakoli et al., 2022; Kajino, 2019), providing multidimensional insights into molecular structures (Nachmani & Wolf, 2020; Chen et al., 2021; Chen & Schwaller, 2023). Despite their promise, these methods still face hurdles in integrating spatial information effectively.

### 2.3 Fragmentation methods

Fragmentation methods break down complex molecules for simpler *ab initio* QM computations of properties, later combining these for a holistic view (Gordon et al., 2012; Collins & Bettens, 2015). Leveraging the localized nature of chemical reactions, these techniques aim for scalable algorithms suitable for large molecule analysis. While instrumental in computational pretraining (Du et al., 2021; Kim et al., 2022b; Luong & Singh, 2023), they typically neglect the fusion of 2D structural with 3D spatial data. Hence, we advocate for a refined fragmentation approach that merges chemical properties with spatial context, potentially advancing hypergraph-based chemical modeling.

## 3 Preliminaries

### 3.1 Equivariance

Consider a function $\mathcal{L}$ that maps inputs from space $\mathcal{X}$ to outputs in space $\mathcal{Y}$. $\mathcal{L}$ is called $G$-equivariant if it preserves the symmetry of a group $G$ across mappings, meaning for each $g \in G$, we have:

$$\mathcal{L} \circ D^{\mathcal{X}}(g) = D^{\mathcal{Y}}(g) \circ \mathcal{L}, \tag{1}$$

where $D^{\mathcal{X}}$ represents the group $G$'s action on $\mathcal{X}$. This ensures that the function $\mathcal{L}$ reflects changes made to inputs by $G$ in its outputs.

### 3.2 Hypergraph

Hypergraphs elegantly capture the essence of higher-order interactions among multiple entities, making them an invaluable tool for representing complex relational data. Let $\mathcal{G} = (V, E)$ be a hypergraph with $N$ vertices and $M$ hyperedges, where $V$ represents a set of nodes and $E$ is a set of hyperedges. Distinguishing itself from a traditional graph, a hyperedge can encompass multiple nodes, not limited to two, i.e. each hyperedge $e \in E$ is a non-empty subset of $V$.

### 3.3 AllSet

The AllSet framework (Chien et al., 2021b), an advanced HGNN model, addresses heuristic propagation rule limitations in HGNNs by integrating Deep Sets (Zaheer et al., 2017) and Set Transformers (Lee et al., 2019) principles. It uses task-optimized dual multiset functions that maintain permutation invariance, crucial for hypergraph learning. The update rules in AllSet are:

$$Z_{e,:}^{(t+1),v} = f_{\mathcal{V} \to \mathcal{E}}(V_{e \setminus v, X^{(t)}}; Z_{e,:}^{(t),v}, X_{v,:}^{(t)}), \tag{2}$$

$$X_{v,:}^{(t+1)} = f_{\mathcal{E} \to \mathcal{V}}(E_{v, Z^{(t+1),v}}; X_{v,:}^{(t)}). \tag{3}$$

Here, $f_{\mathcal{V} \to \mathcal{E}}$ and $f_{\mathcal{E} \to \mathcal{V}}$ are the key multiset functions mapping node and hyperedge features. For example, $f_{\mathcal{V} \to \mathcal{E}}(S) = \text{MLP}\left(\sum_{s \in S} \text{MLP}(s)\right)$ is used in AllDeepSets. The notation $V_{e,X}$ and $E_{v,Z}$ represent multisets of node and hyperedge features, respectively. The AllSet approach updates nodes and hyperedges in the hypergraph by leveraging their features in conjunction with those of adjacent hyperedges or nodes, enabling a rich representation of the hypergraph structure. The method could differentiate node $v$ from its multiset, allowing for sophisticated feature aggregation.

# 4    Methods

We introduce the SE3Set model to leverage hypergraph neural networks for capturing complex molecular interactions, integrating both 2D chemical and 3D spatial structures (Sec. 4.1). It builds upon the AllSet framework (Chien et al., 2021b) and the Equiformer (Liao & Smidt, 2022). Upcoming sections will delve into the specifics of molecular fragmentation and the SE3Set architecture.

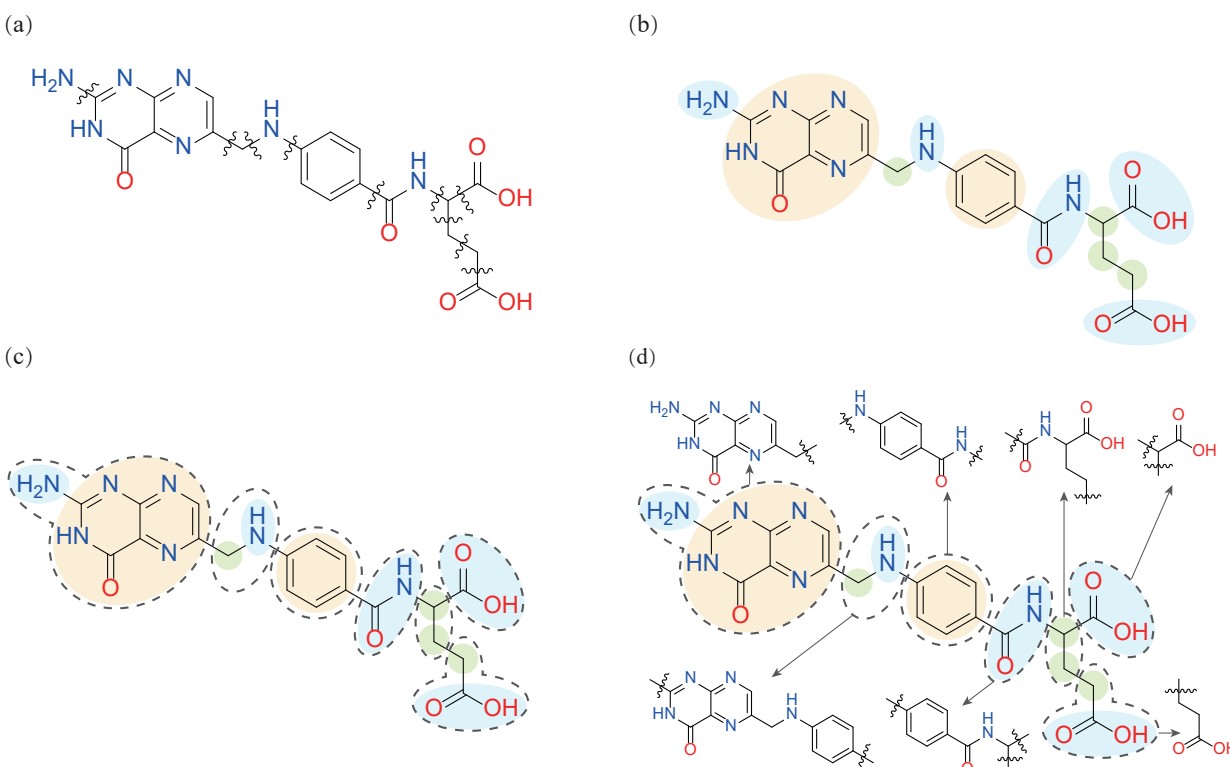

Figure 1: Folic acid fragmentation illustrated with CID 135398658 from PubChem. (a) Preprocessing to identify cleavable bonds for fragmentation. (b) Initial fragments formed using BFS, color-coded by functional groups (blue), rings (orange), and single atoms (green). (c) Fragments merged to satisfy atom count criteria, detailed in D. (d) Expansion of fragments shown with directional arrows.

## 4.1    Fragmentation algorithm

To harness the power of hypergraph neural networks for molecular representations, we need to map molecules onto hypergraph structures through a refined fragmentation algorithm. Our strategy intertwines molecular topology and spatial geometry to create hyperedges that capture groups of atoms, reflecting their functional and spatial characteristics. In crafting this fragmentation approach for hypergraph-based molecular representation, the methodology must adhere to a set of fundamental principles:

1. The design should merge topological chemistry with 3D structural data into a unified hypergraph representation, ensuring hyperedges accurately embody the molecule's chemical and spatial properties.

2. Controlling fragment size is vital for the SE3Set model to balance capturing meaningful interactions and computational efficiency. Optimal fragment sizes are key for model performance and learning capabilities.

3. The fragmentation could only selectively break single bonds and must maintain functional groups and ring integrity to preserving key chemical information critical for the molecule's properties and behavior.

4. Fragment overlap is essential to maintain functional group effects on local charge distribution and to ensure hyperedge interaction within the SE3Set model for improved molecular learning.

Before delving deeper into the specifics of our fragmentation method, it's important to establish a foundational understanding through key definitions and concepts,

**Definition 4.1.** The bond order represents the multiplicity or the number of shared electron pairs that constitute a covalent bond between two atoms.

The bond order matrix $\mathbf{B}$ is an $N \times N$ representation of bond strength between atoms in a molecule, with higher bond order values indicating stronger bonds. This symmetric matrix ($B_{ij} = B_{ji}$) is crucial for studying molecular structure and reactivity, capturing bond nuances including delocalized and resonance bonds in computational chemistry.

Our fragmentation algorithm improves molecular representations by combining bond order, functional groups, and substructures, including SMARTS-identified smaller rings and merged adjacent groups. Overcoming the drawbacks of non-overlapping fragmentation, it uses 3D spatial data and allows overlaps, preserving local effects for precise charge distribution and enhancing hypergraph neural network learning of molecular interactions.

**Definition 4.2.** A molecular fragment, denoted as $\mathfrak{F}$, is defined as a specific subset of atoms within a molecule, characterized by being a cohesive assembly of predefined substructures linked in a sequential concatenation.

Our fragmentation method meticulously dissects a given molecule into meaningful subsets of atoms, and this process unfolds through four steps (corresponding to the pipeline in Fig. 1):

1. Pre-processing by analyzing the molecule's bond order matrix to mask high-order bonds and those within functional groups or rings, and merging adjacent functional groups for a streamlined structural representation.

2. Core substructures are delineated from the remaining bonds using a Breadth-First Search algorithm, establishing the basic units of the molecular framework.

3. These substructures are then aggregated into larger molecular fragments according to predefined rules that maintain a minimum atom count within each fragment. This step could be optional.

4. To enhance fragment connectivity, we expand each by incorporating adjacent groups, using interaction strength metrics based on interatomic distances to guide this process. Here we set the cutoff value denoted as $c_w$ of an interaction strength metrics to intercept the extended fragment.

Furthermore, step 4 leads to a substantial computational overhead for hypergraph neural networks when processing larger molecular systems. To enhance the efficiency of our model for such expansive molecular systems, we introduce a revised strategy for step 4:

4* For each atom $i$, identify the neighboring atoms $\mathcal{N}_i$ that fall within a specified radial cutoff $r_c$. A fragment $\mathfrak{F}$ is considered to be adjacent to atom $i$ if there is an overlap of at least one atom between $\mathfrak{F}$ and $\mathcal{N}_i$. For ease of reference, the set of fragments adjacent to atom $i$ is represented as $\mathcal{N}_i^{\mathcal{F}}$, which implies that $\mathcal{N}_i^{\mathcal{F}} = \{\mathfrak{F} | \mathfrak{F} \cap \mathcal{N}_i \neq \varnothing\}$.

We designate the application of step 4 as an explicit overlap and the application of step 4* as an implicit overlap. These approaches introduce nuanced variations in the mathematical expressions of our model, as reflected in Eq. 8, Eq. 9, and Eq. 16. For the detailed step-by-step methodology, please refer to Appendix B.

## 4.2 SE3Set

Building upon our aforementioned fragmentation algorithm, we now turn to outline the architecture of SE3Set. The SE3Set model, influenced by AllSet (Chien et al., 2021b) and built on the Equiformer (Liao & Smidt,

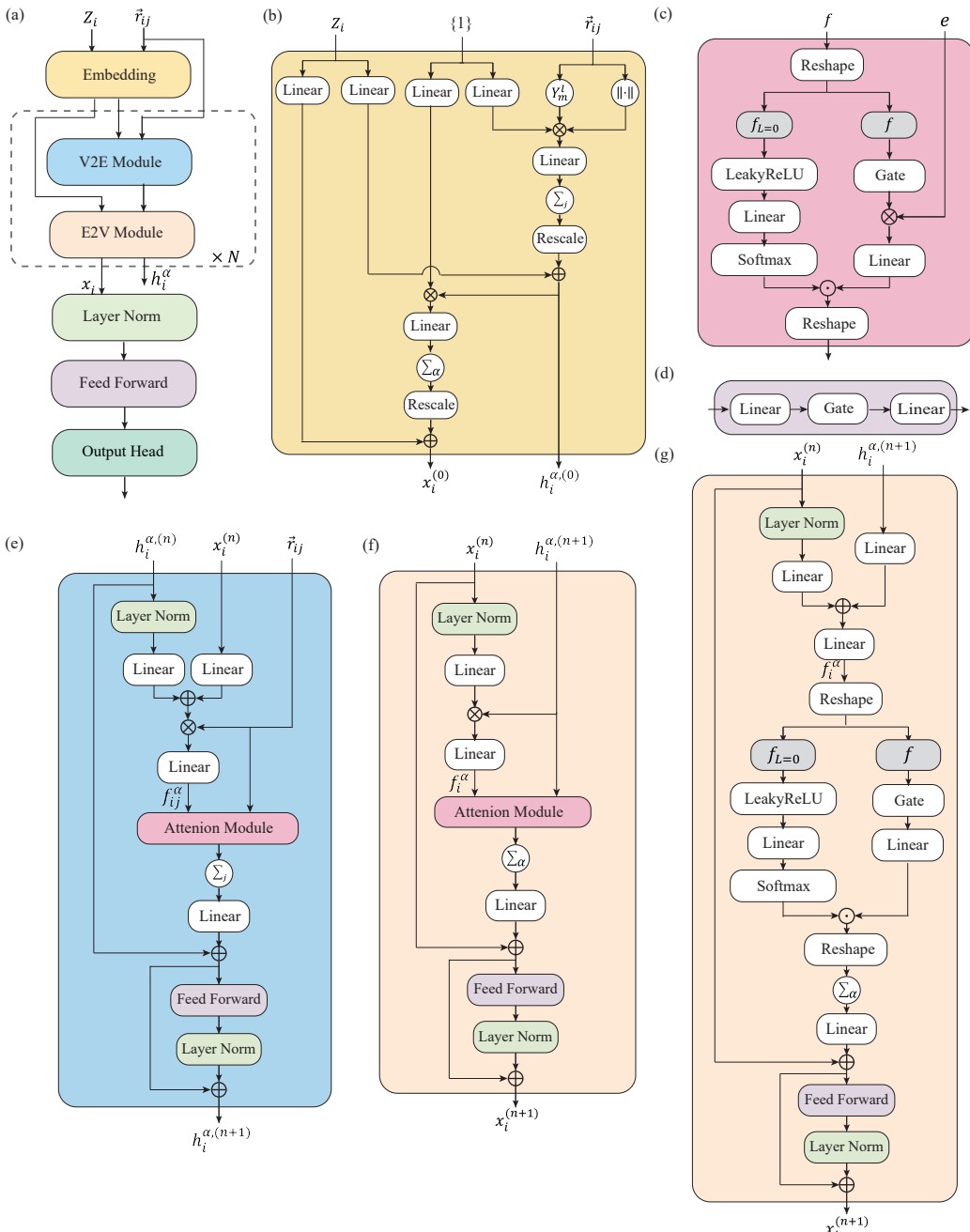

Figure 2: Overall architecture of SE3Set. (a) SE3Set begins with node and hyperedge embeddings, cycles through V2E and E2V attention modules for iterative updates, and concludes with normalization and a feed-forward block for output. (b) Embedding. Atomic numbers and position vectors are transformed into initial embeddings for nodes and hyperedges. (c) Attention Block. Merges feature sets with positional or hyperedge data for feature processing. (d) Feed-Forward Block. Enhances feature sets through a streamlined network. (e) V2E Module. Utilizes node features and their relative positions to update hyperedge features. (f) E2V Module. Employs hyperedge features to refresh node features, using tensor products (left) or summation (right) for updates. Symbols $\otimes$, $\oplus$, and $\odot$ in figures denote depth-wise tensor product, summation, and Hadamard multiplication, respectively. $h_i^\alpha$ represents hyperedge features, $x_i$ is for node features, superscript $n$ indicates the number of updates, and $\vec{r}_{ij}$ is the relative position vector between nodes $i$ and $j$.

2022), incorporates 3D spatial equivariance (proof refers to Appendix F) in our hypergraph neural network, improving the capture of many-body interactions for precise molecular structure representation. SE3Set consists of an embedding layer, attention blocks, and an output head, as shown in Fig. 2 (a). Moreover, from a simple example, we give a brief discussion at Appendix M to the advantages of the hypergraph structure we implement when considering message passing efficiency, comparing to the ordinary graph neural networks.

### 4.2.1 Embedding

As depicted in Fig. 2 (b), the embedding block generates detailed node and hyperedge features reflecting molecular structures. Node features blend intrinsic properties with degree embeddings from connected hyperedges, while hyperedge features aggregate node embeddings and the corresponding relative position vectors depending on the node on which the hyperedge feature is located. SH (Spherical Harmonic) functions are used to project normalized relative position vectors $\vec{r}_{ij}$ between node $i$ and $j$ into the irreducible representations (irreps) feature space with different order $l$, i.e. $\text{SH}(\vec{r}_{ij}) = Y^l\left(\frac{\vec{r}_{ij}}{\|\vec{r}_{ij}\|}\right)$. The features are also mapped onto the same $l$-order irreps space for SE3 equivariance and updated separately. Hyperedges capture nodes' positional relationships, assigning a distinct feature $h_i^\alpha$ to each node $i$ in hyperedge $\mathfrak{F}_\alpha$, reinforcing structural fidelity. Nodes $x_i$ integrate hyperedge information, harmonizing uniqueness with interconnections. Attention mechanisms then refine node and hyperedge interactions for accurate molecular and structural representation.

### 4.2.2 Equivariant hypergraph attention blocks

As presented in Fig. 2 (c)-(f), the attention mechanism comprises two essential components: the Vertex-to-Edge (V2E) and Edge-to-Vertex (E2V) attention blocks, based on the AllSet framework (Chien et al., 2021b). The V2E block refines hyperedge features, while the E2V block updates node features, both operating with an equivariant hypergraph attention mechanism. To improve training and enable deeper network structures, we incorporate normalization layers and residual connections to prevent gradient issues. The attention module's output passes through a feed-forward block (Fig. 2 (d)), enhancing representation complexity. Node and hyperedge features maintain equivariance to molecular geometry, preserving data symmetries and the integrity of representations, thus boosting the model's expressiveness in capturing complex structural interactions. (The concepts of irreducible representations and tensor products can be referenced in the Appendix G.)

**V2E attention** The SE3Set model uses geometrically invariant attention weights $a_{ij}$, derived from $l = 0$ irreps acting as scalars under geometric transformations. These weights are computed from scalar features $f_{ij,l=0}$ using an MLP with LeakyReLU activation and softmax normalization, reflecting node relationships within the hypergraph. Node and hyperedge features undergo non-linear transformations represented by tensor products of irreps with quantum number $l$. The features combine through direct tensor products (DTP), yielding non-linear values $v_{ij}$ (Fig. 2 (c)). Hyperedge features are updated by aggregating features from connected nodes, utilizing SH and radial basis functions on hyperedge features. The model calculates initial features $f_{ij}^\alpha$ and V2E attention weights $a_{ij}^\alpha$ via MLPs, with non-linear values $v_{ij}^\alpha$ emerging from similar transformations.

$$t_{ij}^\alpha = (\text{Linear}(x_i) + \text{Linear}(x_j)) \tag{4}$$

$$f_{ij}^\alpha = \text{Linear}(t_{ij}^\alpha \otimes_{w(\|\vec{r}_{ij}\|)}^{\text{DTP}} \text{SH}(\vec{r}_{ij})) \tag{5}$$

$$a_{ij}^\alpha = \text{Softmax}_j(a^\top \text{LeakyReLU}(f_{ij,l=0}^\alpha)) \tag{6}$$

$$v_{ij}^\alpha = \text{Linear}(\text{Gate}(f_{ij}^\alpha) \otimes_{w(\|\vec{r}_{ij}\|)}^{\text{DTP}} \text{SH}(\vec{r}_{ij})) \tag{7}$$

Ultimately, the SE3Set model updates hyperedge features $h_i^k$ by accumulating the weighted features of nodes within the same hyperedge and applying a linear transformation to the aggregated information. For the explicit overlap fragmentation method,

$$\Delta h_i^\alpha = \text{Linear}\left(\sum_{j:n_i \in \mathfrak{F}_\alpha \wedge n_j \in \mathfrak{F}_\alpha} a_{ij}^\alpha v_{ij}^\alpha\right) \tag{8}$$

where $n_i$ denotes the node with index $i$ and $\mathfrak{F}_\alpha$ denotes the fragment with index $\alpha$ as each fragment could be considered as a hyperedge in the hypergraph. Due to the frequent occurrence of a high number of explicitly overlapping atoms, this scenario commonly results in increased computational complexity. Consequently, when adopting the implicit overlap approach, we may opt for an equation of the form:

$$\Delta h_i^\alpha = \text{Linear} \left( \sum_{j: j \in \mathfrak{F}_\alpha \wedge \mathfrak{F}_\alpha \in \mathcal{N}_i^\mathcal{F}} a_{ij}^\alpha v_{ij}^\alpha \right) \tag{9}$$

where $\mathcal{N}_i^\mathcal{F}$ is delineated in step 4* of the implicit overlap method. This characteristic renders it a more computationally efficient scheme for Vertex-to-Edge (V2E) attention mechanisms. The detailed architecture of the V2E attention block is shown in Fig. 2 (e).

**E2V attention**   Following the V2E attention module, the E2V attention module (Fig. 2 (f)) updates node features by transforming them with a tensor product of the updated hyperedge feature, followed by a linear layer. Attention weights are then calculated using softmax-applied, LeakyReLU-activated features, ensuring node features are refined after hyperedge updates.

$$f_i^\alpha = \text{Linear}((\text{Linear}(x_i) \otimes^{\text{DTP}} h_i^\alpha) \tag{10}$$

$$a_i^\alpha = \text{Softmax}_\alpha(a^\top \text{LeakyReLU}(f_{i,l=0}^\alpha)) \tag{11}$$

These attention weights direct the synthesis of information, culminating in the calculated value:

$$v_i^\alpha = \text{Linear}(\text{Gate}(f_i^\alpha) \otimes^{\text{DTP}} h_i^\alpha). \tag{12}$$

Furthermore, we propose an alternative method for constructing the E2V attention block as shown in Fig. 2 (g).

$$f_i^\alpha = \text{Linear}(\text{Linear}(x_i) + \text{Linear}(h_i^\alpha)) \tag{13}$$

$$a_i^\alpha = \text{Softmax}_\alpha(a^\top \text{LeakyReLU}(f_{i,l=0}^\alpha)) \tag{14}$$

$$v_i^\alpha = \text{Linear}(\text{Gate}(f_i^\alpha)) \tag{15}$$

However, practical experiments reveal that the previous method yields superior results, with detailed findings presented in Sec. 5.3.

Then the node aggregates all the hyperedge features corresponding to itself to update the node feature,

$$\text{Explicit overlap: } \Delta x_i = \text{Linear} \left( \sum_{\alpha: i \in \mathfrak{F}_\alpha} a_i^\alpha v_i^\alpha \right), \tag{16}$$

$$\text{Implicit overlap: } \Delta x_i = \text{Linear} \left( \sum_{\alpha: \mathfrak{F}_\alpha \in \mathcal{N}_i^\mathcal{F}} a_i^\alpha v_i^\alpha \right) \tag{17}$$

### 4.2.3   Output head

The SE3Set model employs node features to generate predictions, using a feed-forward network to transform these features into the target label's irreps dimension. A summation strategy aggregates node features into a single hypergraph-level representation, which is then processed by a linear layer to output the model's final predictions.

## 5   Results

We tested our equivariant hypergraph neural network on QM9 (Ruddigkeit et al., 2012; Ramakrishnan et al., 2014), MD17 (Chmiela et al., 2017) (see Appendix H), MD22 (Chmiela et al., 2023), and OE62 (Stuke

Table 1: A comparative analysis was performed to assess the Mean Absolute Errors (MAEs) on the QM9 dataset when training SE3Set on a configuration comprising 110,000 training samples and 1,000 validation samples. Bolding shows the best model and underlining shows the second best model and the underlining tilde shows third best model. All the baseline results are extracted from the origin papers (Schütt et al., 2017; Gasteiger et al., 2020; Schütt et al., 2021; Coors et al., 2018; Wang et al., 2022; Thölke & De Fabritiis, 2021; Musaelian et al., 2023; Wang et al., 2024; Liao & Smidt, 2022; Wang et al., 2023).

| | UNIT | SCHNET | DIMENET++ | PAINN | SPHERENET | COMENET | ET | ALLEGRO | VISNET | QUINNET | EQUIFORMER | SE3SET |
|---|---|---|---|---|---|---|---|---|---|---|---|---|
| $\mu$ | $D$ | 0.033 | 0.030 | 0.012 | 0.026 | 0.0245 | 0.011 | - | **0.010** | 0.771 | 0.011 | 0.011 |
| $\alpha$ | $a_0^3$ | 0.235 | 0.044 | 0.045 | 0.046 | 0.0452 | 0.059 | - | **0.041** | 0.047 | 0.046 | 0.045 |
| HOMO | meV | 41 | 25 | 20 | 23 | 23 | 20 | - | 17.3 | 20.4 | **15** | **15** |
| LUMO | meV | 34 | 20 | 28 | 18 | 20 | 18 | - | 14.8 | 17.6 | 14 | **13** |
| GAP | meV | 63 | 33 | 46 | 32 | 32 | 36 | - | 31.7 | **28.2** | 30 | 29 |
| $R^2$ | $a_0^2$ | 0.073 | 0.331 | 0.066 | 0.292 | 0.259 | 0.033 | - | **0.030** | 0.194 | 0.251 | 0.197 |
| ZPVE | meV | 1.70 | 1.21 | 1.28 | **1.12** | 1.20 | 1.84 | - | 1.56 | 1.26 | 1.26 | 1.40 |
| $U_0$ | meV | 14 | 6 | 5.85 | 6 | 6.59 | 6.15 | 4.7 | **4.23** | 7.6 | 6.59 | 5.74 |
| $U$ | meV | 19 | 6 | 5.83 | 7 | 6.82 | 6.38 | 4.4 | **4.25** | 8.4 | 6.74 | 5.69 |
| $H$ | meV | 14 | 7 | 5.98 | 6 | 6.86 | 6.16 | **4.4** | 4.52 | 7.8 | 6.63 | 5.70 |
| $G$ | meV | 14 | 8 | 7.35 | 8 | 7.98 | 7.62 | **5.7** | 5.86 | 8.5 | 7.63 | 6.63 |
| $C_v$ | $\frac{\text{kcal}}{\text{mol·K}}$ | 0.033 | 0.023 | 0.024 | **0.021** | 0.024 | 0.026 | - | 0.023 | 0.024 | 0.023 | 0.025 |

et al., 2020) (see Appendix I) to assess its molecular representation learning. QM9 and MD17 gauge small molecule property prediction, while MD22 and OE62 evaluate larger systems with complex many-body interactions (Wang et al., 2023). An ablation study was also conducted to pinpoint the contributions of fragmentation and architecture to our method's performance, offering insights into the network's efficacy and areas for enhancement. In the Appendix J, we have also provided a detailed analysis of the model's complexity. Additionally, we have evaluated the robustness of our model to different tasks by some summary statistic metrics in Appendix N.

## 5.1 QM9

The QM9 dataset (Ruddigkeit et al., 2012; Ramakrishnan et al., 2014) consists of 134k small organic molecules calculated at the B3LYP/6-31G(2df, p) level. SE3Set, after training on 110k QM9 molecules and validation on 10k, achieves low mean absolute errors (MAEs) in 12 tasks, performing on par with leading models, as detailed in Table 1. For completeness, some pre-trained molecular representation learning model are also selected as baselines to give a comparison (see Appendix L). In small molecular systems, higher-order many-body interactions are less pronounced, and as a result, SE3Set does not significantly outperform other state-of-the-art (SOTA) models.

Table 2: A comparison of Mean Absolute Errors (MAEs) across various benchmarked models. SE3Set is trained on the five molecules of MD22 dataset with specific number of training/validation. Bolding shows the best model and underlining shows the second best model. The improvements column shows the improvement of our model over the previous SOTA model in percentage terms. The MAEs reflect the precision of energy predictions in units of kcal/mol and forces in units of kcal/(mol·Å). The results of the baseline models refs to the origin papers (Chmiela et al., 2023; Thölke & De Fabritiis, 2021; Musaelian et al., 2023; Batatia et al., 2022; Liao & Smidt, 2022; Wang et al., 2024; 2023; Li et al., 2024).

| MOLECULE | # TRAIN/VAL | | sGDML | TORCHMD-NET | ALLEGRO | MACE | EQUIFORMER | VISNET | QUINNET | EQUIFORMER-LSRM | VISNET-LSRM | SE3SET | IMPROVEMENTS |
|---|---|---|---|---|---|---|---|---|---|---|---|---|---|
| Ac-Ala3-NHMe | 5500/500 | ENERGY | 0.3902 | 0.1121 | 0.1019 | 0.0620 | 0.0828 | 0.0796 | 0.084 | 0.0780 | 0.0654 | **0.0499** | 19.5% |
| | | FORCE | 0.7968 | 0.1879 | 0.1068 | 0.0876 | 0.0804 | 0.0972 | 0.0681 | 0.0877 | 0.0902 | **0.0545** | 20.0% |
| DHA | 7500/500 | ENERGY | 1.3117 | 0.1205 | 0.1153 | 0.1317 | 0.1788 | 0.1526 | 0.12 | 0.0878 | 0.0873 | **0.0826** | 5.4% |
| | | FORCE | 0.7474 | 0.1209 | 0.0732 | 0.0646 | 0.0506 | 0.0668 | 0.0515 | 0.0534 | 0.0598 | **0.0360** | 28.9% |
| STACHYOSE | 7500/500 | ENERGY | 4.0497 | 0.1393 | 0.2485 | 0.1244 | 0.1404 | 0.1283 | 0.23 | 0.1252 | 0.1055 | **0.0762** | 27.8% |
| | | FORCE | 0.6744 | 0.1921 | 0.0971 | 0.0876 | 0.0635 | 0.0869 | 0.0543 | 0.0632 | 0.0767 | **0.0424** | 21.9% |
| AT-AT | 2500/500 | ENERGY | 0.7235 | 0.1120 | 0.1428 | 0.1093 | 0.1309 | 0.1688 | 0.14 | 0.1007 | 0.0772 | **0.0585** | 24.2% |
| | | FORCE | 0.6911 | 0.2036 | 0.0952 | 0.0992 | 0.0960 | 0.1070 | 0.0687 | 0.0811 | 0.0781 | **0.0556** | 19.1% |
| AT-AT-CG-CG | 1500/500 | ENERGY | 1.3885 | 0.2072 | 0.3933 | 0.1578 | 0.1510 | 0.1995 | 0.38 | 0.1335 | 0.1135 | **0.1002** | 11.7% |
| | | FORCE | 0.7028 | 0.3259 | 0.1280 | 0.1153 | 0.1252 | 0.1563 | 0.1273 | 0.1065 | 0.1063 | **0.0825** | 22.4% |

Table 3: The MAE for energy (unit: kcal/mol) and forces (unit: kcal/mol · Å) on the AT-AT-CG-CG dataset using 3 layers SE3Set with different cutoff radii ($r_c$) in the implicit overlap method.

| Implicit $r_c$(Å) | 4.0 | 5.0 | 6.0 |
|---|---|---|---|
| Energy | 0.2123 | 0.1153 | 0.1103 |
| Force | 0.1559 | 0.1019 | 0.0937 |

## 5.2 MD22

Recognizing the prominence of higher-order many-body interactions in larger molecules (Wang et al., 2023), SE3Set was tested on the comprehensive MD22 dataset (Chmiela et al., 2023). This dataset spans four classes of biomolecules and supramolecules, from a 42-atom peptide to a 370-atom nanotube, with high-resolution sampling at 400-500 K using the PBE+MBD (Perdew et al., 1996; Tkatchenko et al., 2012) framework for energy and force computations. Our fragmentation method, which maintains functional groups and rings, selectively excludes structures like the Buckyball catcher and Double-walled nanotube from MD22, thus concentrating on the other five molecular types. We partition the training/test set following QuinNet (Wang et al., 2023). As Table 2 shows, SE3Set outperforms other SOTA models in these cases, reducing MAEs by an average of 20%, underscoring its exceptional ability to capture molecular intricacies. Moreover, our results indicate that incorporating higher-order many-body interactions is crucial for representing the non-local features of larger molecules within the MD22 dataset.

## 5.3 Ablation studies

To better understand SE3Set, we conduct ablation studies focusing on fragmentation and model architecture. For the explicit overlap fragmentation method, we explore how different fragmentation techniques affect SE3Set's training and compare with the non-overlapping BRICS (Degen et al., 2008; Landrum et al., 2020) strategy on QM9's homo energy task. As Fig. 3 indicates, tests on QM9's homo energy task showed SE3Set's robustness to $c_w$ variations in fragmentation method. The results show that our method surpasses BRICS, demonstrating the importance of hyperedge interaction. Furthermore, we performed ablation studies on the model architecture. Among two design variants in the E2V attention section, the one using

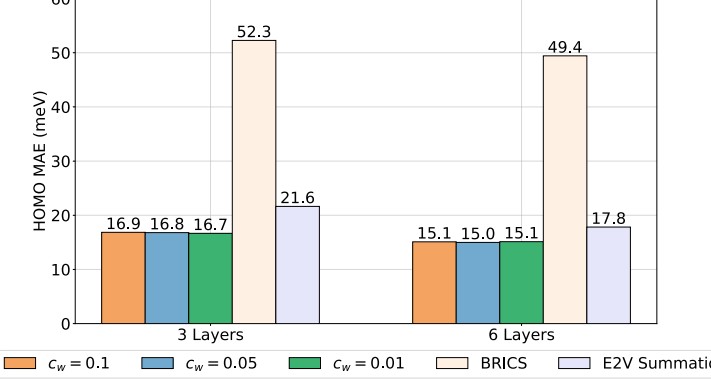

Figure 3: Ablation studies on the QM9 dataset's HOMO task (units: meV). The variable $c_w$ represents the threshold for expansion in the fourth step of fragmentation, guided by the fragment bond order defined in Eq. 18. The term BRICS denotes another fragmentation method implemented in RDKits. Additionally, the E2V summation refers to the architectural framework specified from Eq. 13 to Eq. 15.

tensor product interactions between nodes and hyperedges proved superior, emphasizing the value of our tensor product-based mechanism and architecture design in enhancing molecular property predictions. Additionally, a 6-layer SE3Set model outperformed its 3-layer counterpart.

Unlike the explicit overlap method, the fragment size of the implicit overlap method depends on the choice of $r_c$. We test the effect of $r_c$ on the model training results on the AT-AT-CG-CG molecule of the MD22 dataset (Table 3). The SE3Set performs better when using higher $r_c$ as it will include more fragments to generate implicit overlaps. The effect of $r_c$ on fragment size is more pronounced for the implicit method than for the explicit method, but it gives good performance for different parameters compared with the baseline models.

## 6 Conclusion

In conclusion, this study demonstrates the efficacy of SE3Set, a cutting-edge hypergraph neural network architecture, in the realm of molecular representation learning. By meticulously crafting a fragmentation method that coalesces two-dimensional chemical knowledge with three-dimensional spatial information, we establish a robust foundation for constructing hypergraphs that faithfully capture the complex nature of molecular structures. The SE3Set architecture, drawing inspiration from the AllSet framework and the Equiformer, adeptly processes these hypergraphs and preserves the essential invariances and symmetries. SE3Set demonstrates performance comparable to SOTA models in small molecular systems and significantly outperforms SOTA models in large molecular systems where higher-order many-body interactions are pronounced. The results of our research affirm the potential of SE3Set to model high-order many-body interactions, providing a powerful tool for molecular representation.

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

## A  Appendix

## B  Details of fragmentation steps

Based on the design principles in Sec. 4.1, the detailed step-by-step methodology of explicit overlap fragmentation method is shown as follows,

1. The pre-processing step begins by analyzing the given molecule through its bond order matrix, denoted as $B$. Identify and mask bonds that are part of functional groups or rings, as well as those with a bond order of $B_{ij} \geqslant 2$. Functional groups are then identified using predefined SMARTS patterns for accurate matching. To achieve a more generalized representation of functional groups, topologically adjacent functional groups are merged into a single entity. This aggregation allows to focus on specific subfunctional groups that are of particular interest, simplifying the complexity of the molecular structure for subsequent analysis.

2. Following the masking of selected bonds, the Breadth-First Search (BFS) algorithm is employed to reconstruct the substructures, denoted as $\{\mathfrak{S}\}$, from the remaining unmasked bonds. These groups represent the core structural units of the molecule as discussed at the outset of this section.

3. Consolidate the previously identified groups $\{\mathfrak{S}\}$ into larger molecular fragments, applying a set of predefined rules to guide the merging process. These rules are meticulously designed to ensure that each resulting fragment, now denoted as $\{\mathfrak{F}\}$, contains at least a minimum specified number of atoms. For a comprehensive understanding of the merging criteria, one can refer to the detailed rules outlined in D.

4. Extend each fragment $\{\mathfrak{F}\}$ by incorporating adjacent groups from $\{\mathfrak{S}\}$ to enrich the connectivity between molecular fragments, thus intentionally creating regions of overlap among the fragments. This expansion is controlled by a cutoff threshold, denoted as $c_w$, which is typically a function based on interatomic distances. the fragment bond order (Lendvay, 2000; Bridgeman & Empson, 2006), symbolized by $W_{fs}$, is used to quantitatively assess the interaction strength between a fragment $\mathfrak{F}_i$ and an adjacent substructure $\mathfrak{S}_j$. This method reflects the interaction strength based on the proximity of atoms in different fragments, expressed by the following equation:

$$W_{fs} = \sum_{i \in \mathfrak{F}_f, j \in \mathfrak{S}_s} \exp\left(-\frac{(d_{ij} - d_{ij}^e) \cdot d_{ij}^e}{(0.25 \text{ Å})^2}\right), \tag{18}$$

where $d_{ij}$ represents the interatomic distance between atoms $i$ and $j$, and $d_{ij}^e$ stands for the equilibrium distance typically expected for such a bond. This equation is utilized to determine which substructures should be included in the expansion of a fragment, based on the strength of their interactions as governed by the distance function. Additionally, in alignment with Pauling's concept of "chemist's bond order" (Lendvay, 2000), an alternative method is introduced to calculate the bond order using a single exponential function,

$$W_{fs} = \sum_{i \in \mathfrak{F}_f, j \in \mathfrak{S}_s} \exp\left(-(d_{ij} - d_{ij}^e)\right), \tag{19}$$

where $W_{fs}$ encapsulates the bond order between atoms belonging to a fragment $\mathfrak{F}_f$ and a substructure $\mathfrak{S}_s$. In this context, $d_{ij}$ signifies the actual measured distance between atom $i$ and atom $j$. The term $d_{ij}^e$ refers to the theoretical equilibrium covalent bond length, which is estimated by summing the empirical covalent radii of the two atoms involved, given by:

$$d_{ij}^e = r_{z_i} + r_{z_j}, \tag{20}$$

where $r_{z_i}$ is the empirical covalent radius of an atom with atomic number $z_i$. This function provides a simplified yet effective representation of bond order, allowing us to gauge the bonding interactions within the molecular structure with respect to the proximity of the atoms.

The implicit overlap fragmentation method only changes the step 4, the details of the changed fourth step have been spelled out in 4.1.

## C Functional groups SMARTS

In the initial phase of our fragmentation approach, we identify functional groups using the SMARTS pattern matching language. In Table 4, we present the complete list of SMARTS patterns utilized, which have been expanded upon from the default set found within the Open Force Field toolkit (Mobley et al., 2018; Wagner et al., 2024) (accessible at: `https://github.com/openforcefield/openff-fragmenter/blob/main/openff/fragmenter/data/default-functional-groups.json`).

Table 4: SMARTS patterns for functional groups employed in the preprocessing stage of fragmentation.

| FUNCTIONAL GROUPS NAME | SMARTS |
| --- | --- |
| HYDRAZINE | [NX3:1][NX3:2] |
| HYDRAZONE | [NX3:1][NX2:2] |
| NITRIC OXIDE | [N:1]-[O:2] |
| AMIDE | [#7:1][#6:2](=[#8:3]), [NX3:1][CX3:2](=[OX1:3])[NX3:4] |
| AMIDE NEGATIVE ION | [#7:1][#6:2](-[O-:3]) |
| ALDEHYDE | [CX3H1:1](=[O:2])[#6:3] |
| SULFOXIDE | [#16X3:1]=[OX1:2], [#16X3+:1][OX1-:2] |
| SULFONYL | [#16X4:1](=[OX1:2])=[OX1:3] |
| SULFINIC ACID | [#16X3:1](=[OX1:2])[OX2H,OX1H0-:3] |
| SULFONIC ACID | [#16X4:1](=[OX1:2])(=[OX1:3])[OX2H,OX1H0-:4] |
| SULFINAMIDE | [#16X4:1](=[OX1:2])(=[OX1:3])([NX3R0:4]) |
| PHOSPHINE OXIDE | [PX4:1](=[OX1:2])([#6:3])([#6:4])([#6:5]) |
| PHOSPHONATE | [P:1](=[OX1:2])([OX2H,OX1-:3])([OX2H,OX1-:4]) |
| PHOSPHATE | [PX4:1](=[OX1:2])([#8:3])([#8:4])([#8:5]) |
| CARBOXYLIC ACID | [CX3:1](=[O:2])[OX1H0-,OX2H1:3] |
| NITRO | [NX3+:1](=[O:2])[O-:3], [NX3:1](=[O:2])=[O:3] |
| ESTER | [CX3:1](=[O:2])[OX2H0:3] |
| TRI-HALIDE | [#6:1]([F,Cl,I,Br:2])([F,Cl,I,Br:3])([F,Cl,I,Br:4]) |
| HYDROXYL | [#8:1]-[#1:2] |

## D Merge process of fragmentation

During the third step of our fragmentation method, we introduce a strategy to enlarge substructures, ensuring that each initial fragment contains at least $n_{\min}$ atoms, with $n_{\min}$ being a predefined integer. To maintain permutation invariance for a molecule, we incorporate weights, $W_{fs}$, to guide the sequence of merging. The process is outlined in the pseudocode (Algorithm 1). The calculation of $W$ is based on either Eq. 18 or Eq. 19. By considering the sum of bond orders to other groups, we assess each group's centrality. The groups are then ordered first by the number of atoms they contain, followed by the summation of their bond orders, ensuring that the fragmentation merge process is permutation invariant when following this specified sequence. The algorithm then assists smaller groups in merging with others to achieve a size of at least $n_{\min}$ atoms. Initially, we consider topologically adjacent groups with the fewest atoms. If a target group lacks topological neighbors, we proceed to merge based on the bond order from $W$. We introduce a threshold $c_{is}$ that allows a group to remain isolated if it is significantly distant from others. It should be noted that isolated groups may not meet the minimum atom number requirement; however, they could be further expanded in the subsequent fragmentation step, depending on the chosen thresholds for $c_{is}$ and $c_w$ (refer to Sec. 4.1). Overall, this algorithm ensures a permutation invariant merging process.

## E Distribution of fragmentation dataset

Different parameters used in the fragmentation process can lead to a variety of hyperedges, which in turn result in distinct hypergraphs utilized for training our model. To illustrate the variances attributed to different fragmentation parameters or methods (such as BRICS implemented in RDKit (Degen et al., 2008; Landrum

---

**Algorithm 1** Pseudo code of fragmentation merge step.

---

**Input:** groups $\{\mathfrak{G}\}$, minimum atoms number $n_{\min}$, maximum atoms number $n_{\max}$, Topological bond order matrix $B$, isolated threshold $c_{is}$
$m = |\{\mathfrak{G}\}|$
Isolate groups $\{\mathfrak{G}^{\mathfrak{I}}\} = \{\}$
Calculate fragmentation bond order matrix $W_{\mathfrak{G}_i \mathfrak{G}_j}$.
Sort $\{\mathfrak{G}\}$ in descending order based on the following attributes: number of atoms, $\sum_{\mathfrak{G}', \mathfrak{G}' \neq \mathfrak{G}} W_{\mathfrak{G} \mathfrak{G}'}$.
**repeat**
    Pop last fragment as $\mathfrak{G}_k$ from $\{\mathfrak{G}\}$
    **for** $i = m - 1$ **to** 1 **do**
        $a = \text{MAX\_INT}, merge\_idx = -1$
        **if any** $B_{ij} \geqslant 1, i \in \mathfrak{G}_i j \in \mathfrak{G}_k$ **and** $|\mathfrak{G}_i| < a$ **and** $a + |\mathfrak{G}_i| \leqslant n_{\max}$ **then**
            $a = |\mathfrak{G}_i|, merge\_idx = i$
        **end if**
    **end for**
    **if** $merge\_idx == -1$ **then**
        **for** $i = m - 1$ **to** 1 **do**
            **if any** $W_{\mathfrak{G}_i \mathfrak{G}_k} \geqslant c_{is}$ **and** $|\mathfrak{G}_i| < a$ **then**
                $a = |\mathfrak{G}_i|, merge\_idx = -1$
            **end if**
        **end for**
    **end if**
    **if** $merge\_idx \neq -1$ **then**
        Merge $\mathfrak{G}_k$ to $\mathfrak{G}_{merge\_idx}$
        Resort $\{\mathfrak{G}\}$ by the same priority and update $W$.
    **else**
        Add $\{\mathfrak{G}_k\}$ to $\{\mathfrak{G}^{\mathfrak{I}}\}$
    **end if**
**until** $|\{\mathfrak{G}_k\}| \geqslant n_{\min}$
$\{\mathfrak{F}\} = \{\mathfrak{G}\} \cup \{\mathfrak{G}^{\mathfrak{I}}\}$

---

et al., 2020)), we use the QM9 dataset (Ruddigkeit et al., 2012; Ramakrishnan et al., 2014) to demonstrate how the data distributions attached to hypergraphs may change.

The impact of adjusting fragmentation parameters on the composition of hyperedges can be observed in Fig. 4. Altering the expansion threshold $c_w$ within a certain range has a minimal effect on fragment expansion. However, when utilizing the Lendvay bond order (Eq. 18), fragments tend to comprise fewer atoms compared to when using the Exponential bond order (Eq. 19). This difference is likely due to the more gradual decline in the exponential function, which results in a greater cumulative contribution to the weights $W_{fs}$.

Our ablation study (Sec. 5.3) also includes a comparison with the BRICS fragmentation method. Fragments generated by the BRICS method are observed to contain significantly fewer atoms since this approach does not create overlapping regions between different fragments.

## F   Proof of SE(3) equivariance

The SE3Set consists of the basic modules including linear, depth-wise tensor product, gate activation and layer normalization. Here we will prove the SE3 equivariance for these modules. As the inputs of SE3Set are all invariant to translation in 3D Euclidean space, we only need to prove the $SO(3)$ equivariance.

Let $g \in SO(3)$ and the $\mathcal{D}$ denote the group representation of $SO(3)$. Besides, we denote the irreps feature as $\mathbf{f}$, and $\mathbf{f}^l$ denotes the $k^{\text{th}}$ feature vector in irreducible representations space of $SO(3)$ with $l$ order.

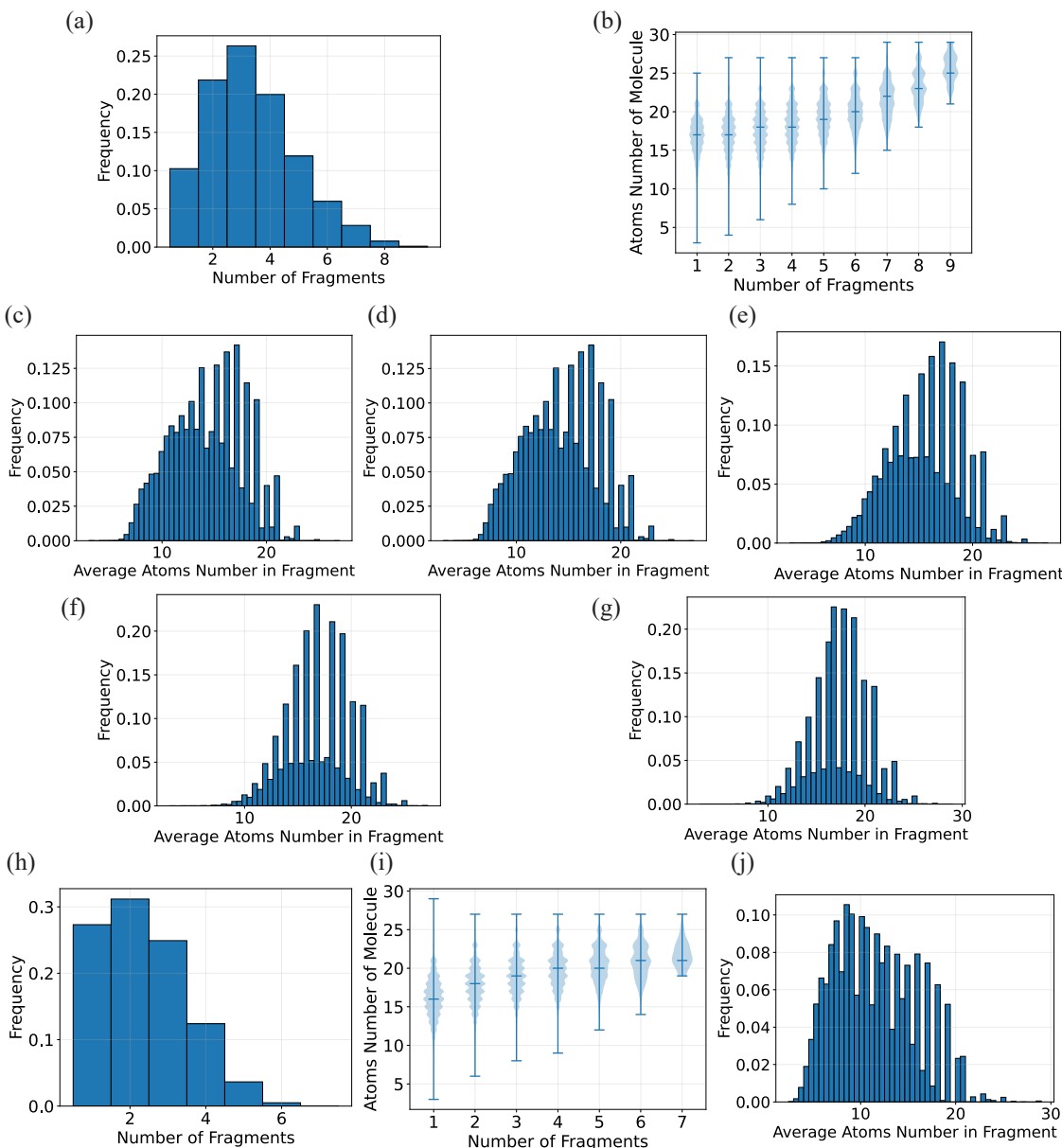

Figure 4: Distribution of fragments in QM9 dataset. (a) Fragment Count Distribution. The distribution remains consistent regardless of the value of $c_w$ or the bond order calculation method employed. (b) Molecule Size vs. Fragment Count Distribution. Generally, the more atoms molecule has, the more fragments will generate. It is also invariant for $c_w$ or bond order calculation scheme. Average Atom Count per Fragment Distribution (c) $c_w = 0.1$, (d) $c_w = 0.05$, (e) $c_w = 0.01$ for Lendvay bond order and (f) $c_w = 0.4$ and (g) $c_w = 0.2$ for exponential bond order, respectively. (h) BRICS Fragment Count Distribution. (i) BRICS Molecule Size vs. Fragment Count Distribution (j) BRICS Average Atom Count per Fragment.

**Linear** Linear module deploys separated linear operations for each $l$ in the irreps feature. For each $l$, we consider the output channel $\mathbf{f}_k^l$. Then we have

$$\mathbf{f}_j^l = \text{Linear}(\mathbf{f}^l) = \sum_k w_{kj}^l \mathbf{f}_k^l \tag{21}$$

where $w_{kj}^l$ denotes the linear combination weight. When acting $\mathcal{D}(g)$ at the input $f^l$, we can find that

$$(\mathbf{f}_j^l)' = \text{Linear}(\mathcal{D}(g)\mathbf{f}^l) \tag{22}$$

$$= \sum_k w_{kj}^l \mathcal{D}(g)\mathbf{f}_k^l \tag{23}$$

$$= \mathcal{D}(g) \sum_k w_{kj}^l \mathbf{f}_k^l \tag{24}$$

$$= \mathcal{D}(g)\mathbf{f}_j^l \tag{25}$$

Therefore, the linear module is $SO(3)$ equivariant.

**Depth-wise Tensor-product** Tensor product is an equivariant operation for $\mathcal{D}(g)$

$$\mathcal{D}(g)(\mathbf{f}_{k_1}^l \otimes \mathbf{f}_{k_2}^l) = (\mathcal{D}(g)\mathbf{f}_{k_1}^l) \otimes (\mathcal{D}(g)\mathbf{f}_{k_2}^l) \tag{26}$$

The depth-wise tensor product differs from the tensor product only in that one order $l$ vector in out irreps feature depends only on one order $l'$ feature, where $l'$ is equal to or different from $l$. Hence the $SO(3)$ equivariance still holds for depth-wise tensor product.

**Gate** As the $l = 0$ vector is invariant for $\mathcal{D}(g)$, we have

$$\text{Gate}(\mathcal{D}(g)\mathbf{f}_k^0) = \text{Activation}(\mathcal{D}(g)\mathbf{f}_k^0) \tag{27}$$

$$= \text{Activation}(\mathbf{f}_k^0) \tag{28}$$

$$= \mathcal{D}(g)\text{Activation}(\mathbf{f}_k^0) \tag{29}$$

$$= \mathcal{D}(g)\text{Gate}(\mathbf{f}_k^0) \tag{30}$$

We use the non-linear output from the $\text{Activation}(\mathcal{D}(g)\mathbf{f}_{k_1}^0)$ (i.e. Sigmoid in SE3Set) as the weight which multiplies $l > 0$ vector to implement the gate function. Thus for $l > 0$, we also have

$$\text{Gate}(\mathcal{D}(g)\mathbf{f}_{k_1}^l) = \text{Activation}(\mathcal{D}(g)\mathbf{f}_{k'}^0)\mathcal{D}(g)\mathbf{f}_{k_1}^l \tag{31}$$

$$= \text{Activation}(\mathbf{f}_{k_1}^0)\mathcal{D}(g)\mathbf{f}_{k_1}^l \tag{32}$$

$$= \mathcal{D}(g)\text{Gate}(\mathbf{f}_{k_1}^l) \tag{33}$$

So the gate function is $SO(3)$ equivariant for all $l$ vector.

**Layer Normalization** For $l = 0$, as it is $SO(3)$ invariant, the general layer normalization is adapted

$$\text{LN}(\mathbf{f}_k^0) = \left(\frac{\mathbf{f}_k^0 - \mu}{\text{RMS}_C(\|\mathbf{f}^0\|)}\right)\gamma + \beta \tag{34}$$

where $C$ denotes the channel number corresponding and $\|\cdot\|$ denotes the 2-norm for each channel vector. $\mu$ is the mean value of $\mathbf{f}^0$. Learnable weight $\gamma$ and learnable bias $\beta$ are also deployed. The module is $SO(3)$ invariant as $\mathbf{f}_k^0$ is $SO(3)$ invariant. The layer normalization for $l > 0$ vector has the following form

$$\text{LN}(\mathbf{f}_k^l) = \left(\frac{\mathbf{f}_k^l}{\text{RMS}_C(\|\mathbf{f}^l\|)}\right)\gamma \tag{35}$$

We can prove the module is $SO(3)$ equivariant as

$$\text{LN}(\mathcal{D}(g)\mathbf{f}_k^l) = \left(\frac{\mathcal{D}(g)\mathbf{f}_k^l}{\text{RMS}_C(\|\mathcal{D}(g)\mathbf{f}^l\|)}\right)\gamma \tag{36}$$

$$= \left(\frac{\mathcal{D}(g)\mathbf{f}_k^l}{\text{RMS}_C(\|\mathbf{f}^l\|)}\right)\gamma \tag{37}$$

$$= \mathcal{D}(g)\text{LN}(\mathbf{f}_k^l) \tag{38}$$

## G   Concepts of irreps features and tensor product

**Irreps features**   The SE3Set model utilizes the special orthogonal group SO(3) to capture three-dimensional rotational symmetries in molecular structures. This approach is similar to Equiformer (Liao & Smidt, 2022; Liao et al., 2023).It employs irreducible representations (irreps) of SO(3), parameterized by an integer $l$, which correspond to spherical harmonics (SH) functions $Y_l^m$. These functions imbue feature vectors with rotational information, ensuring the model's equivariance to rotations and enabling consistent geometric property analysis. This approach is key to the model's ability to accurately represent and predict molecular and other rotationally invariant systems.

**Tensor product**   To boost the model's expressive power, we consider interactions between irrep features of different angular momenta $l$ through the tensor product, which merges two irreps $l_1$ and $l_2$ into a new irrep with angular momentum $l_3$. This is achieved using Clebsch-Gordan coefficients in an expansion weighted by $w_{m_1,m_2}$.

$$
\begin{aligned}
f_{m_3}^{l_3} &= (f_{m_1}^{l_1} \otimes f_{m_2}^{l_2})_{m_3} \\
&= \sum_{m_1,m_2} w_{m_1,m_2} C_{l_1,m_1,\ l_2,m_2}^{l_3,m_3} f_{m_1}^{l_1} f_{m_2}^{l_2}.
\end{aligned}
\tag{39}
$$

To reduce complexity, a depth-wise tensor product $\otimes^{\mathrm{DTP}}$ is adopted from the Equiformer (Liao & Smidt, 2022; Liao et al., 2023), utilizing internal weights to streamline computations. Input-dependent tensor product weights are denoted as $\otimes_w^{\mathrm{DTP}}$, ensuring computational efficiency while preserving equivariance for feature interactions.

## H   Results of MD17

The MD17 dataset (Chmiela et al., 2017) features a wide variety of molecular configurations simulated at 500 K, with high-resolution trajectories and labeled with energies and forces from the PBE+vdW-TS method (Perdew et al., 1996; Tkatchenko et al., 2012). SE3Set's performance on this dataset is shown in Table 5. SE3Set outperforms Equiformer in accuracy, highlighting its refined force calculation capabilities. In small molecular systems, higher-order many-body interactions are less pronounced, and as a result, SE3Set does not significantly outperform other state-of-the-art (SOTA) models.

## I   Results of OE62

The OE62 dataset (Stuke et al., 2020) contains about 62k large organic molecules with annotated DFT-computed energies in the unit of eV. We randomly selected 50000 data points as training set and 6000 data points as training set. Then we report the MAE of energy as the performance metric on the test dataset. Since OE62 is considered to have significant long-range interactions Kosmala et al. (2023), we selected model results without explicit long-range modeling as a baseline. The results (Table 6) shows that SE3Set outperforms all the baseline models (short-range models), which indicates the potential of the many body interaction modeling by our model.

## J   Complexity Analysis

The computational complexity mainly depends on the V2E module and the E2V module.

The V2E module aggregates the information from each atom in one fragment to generate atom-wise hyperedge features. Considering the system is split into $m$ fragments and each fragment has $n_i$ atoms, this module includes $\sum_i^m n_i(n_i - 1)$ pair-wise messages for the attention architecture. Actually, this number depends on the fragment hyperparameter $n_{min}$ and $n_{max}$, in particular for the explicit overlap method, also on $c_w$. For the explicit overlap method, this module would have higher complexity than the implicit overlap method because the explicit overlap has a higher average number of atoms in each fragment.

Table 5: A comparison of Mean Absolute Errors (MAEs) across various benchmarked models. SE3Set is trained on the MD17 dataset with a configuration of 950 training samples and 50 validation samples. Bolding shows the best model and underlining shows the second best model and the underlining tilde shows third best model. The MAEs reflect the precision of energy predictions in units of kcal/mol and forces in units of kcal/(mol·Å). The results of each baseline come from the corresponding articles (Schütt et al., 2017; Gasteiger et al., 2019; Schütt et al., 2021; Thölke & De Fabritiis, 2021; Gasteiger et al., 2021; Batzner et al., 2022; Wang et al., 2024; Liao & Smidt, 2022; Wang et al., 2023).

| | | SchNet | DimeNet | PaiNN | ET | GemNet | NequIP ($l=3$) | ViSNet | QuinNet | Equiformer | SE3Set |
|---|---|---|---|---|---|---|---|---|---|---|---|
| Aspirin | Energy | 0.37 | 0.204 | 0.167 | 0.123 | - | 0.131 | **0.116** | 0.119 | 0.122 | 0.130 |
| | Force | 1.35 | 0.499 | 0.338 | 0.253 | 0.217 | 0.184 | 0.155 | **0.145** | 0.152 | 0.153 |
| Ethanol | Energy | 0.08 | 0.064 | 0.064 | 0.052 | - | 0.051 | 0.051 | **0.050** | 0.051 | 0.054 |
| | Force | 0.39 | 0.230 | 0.224 | 0.109 | 0.085 | 0.071 | **0.060** | **0.060** | 0.067 | 0.062 |
| Malonaldehyde | Energy | 0.13 | 0.104 | 0.091 | 0.077 | - | 0.076 | 0.075 | 0.078 | **0.074** | **0.074** |
| | Force | 0.66 | 0.383 | 0.319 | 0.169 | 0.155 | 0.129 | 0.100 | **0.097** | 0.125 | 0.103 |
| Naphthalene | Energy | 0.16 | 0.122 | 0.116 | 0.085 | - | 0.113 | **0.085** | 0.101 | **0.085** | 0.113 |
| | Force | 0.58 | 0.215 | 0.077 | 0.061 | 0.051 | **0.039** | **0.039** | **0.039** | 0.046 | **0.039** |
| Salicylic acid | Energy | 0.20 | 0.134 | 0.116 | 0.093 | - | 0.106 | **0.092** | 0.101 | 0.099 | 0.108 |
| | Force | 0.85 | 0.374 | 0.195 | 0.129 | 0.125 | 0.090 | 0.084 | **0.080** | 0.090 | 0.090 |
| Toluene | Energy | 0.12 | 0.102 | 0.095 | 0.074 | - | 0.092 | **0.074** | 0.080 | 0.085 | 0.093 |
| | Force | 0.57 | 0.216 | 0.094 | 0.067 | 0.060 | 0.046 | **0.039** | **0.039** | 0.048 | 0.046 |
| Uracil | Energy | 0.14 | 0.115 | 0.106 | 0.095 | - | 0.104 | **0.095** | 0.096 | 0.099 | 0.103 |
| | Force | 0.56 | 0.301 | 0.139 | 0.095 | 0.097 | 0.076 | **0.062** | **0.062** | 0.076 | 0.067 |

Table 6: A comparison of MAEs across some benchmarked models on OE62 dataset. The best model is bolded. The results of baselines are extracted from (Kosmala et al., 2023).

| Model | SchNet | PaiNN | DimeNet++ | GemNet-T | SE3Set(3 Layers) |
|---|---|---|---|---|---|
| MAE (meV) | 131.3 | 63 | 53.8 | 53.1 | **51.7** |

For each atom, the E2V module aggregates all the hyperedge features that an atom possesses. Thus the calculated pair-wise message number depends on the number of fragments per atom shared, corresponding to the introduced overlap degrees. Then the computational complexity of the explicit overlap method in this module depends on the $c_w$ for the explicit overlap method and $r_c$ for the implicit overlap method.

To give a general approximate complexity, we consider a system with $N$ atoms. When using the explicit overlap method, if the system has $m$ fragments with an average $n_{exp}$ atoms in one fragment, the V2E module's complexity will be $O(mn_{exp}^2)$ according to the previous analysis. For the E2V module, the number of fragments to which each atom belongs on average can be represented as $mn_{exp}/N$. Thus, the complexity will be $O(mn_{exp})$ when considering $N$ atoms.

Similarly, for the implicit overlap method, the complexity of the V2E module will be $O(n_{\mathcal{F}}n_{imp}N)$, where $n_{\mathcal{F}}$ represents the average number of neighbor fragments within cutoff per atom and $n_{imp}$ denotes the average atoms in one fragment. And the complexity of the E2V module is $O(n_{\mathcal{F}}N)$.

Compare to the explicit overlap method, the implicit overlap method has lower computational complexity. As we can approximate that $n_{\mathcal{F}}n_{imp} \approx n_{exp}$ (overlap degree is similar) and determine $mn_{imp} = N$ (implicit overlap method has same fragment number $m$ but without extra count of atoms in $n_{imp}$). On the other hand we can determine $mn_{exp} \geqslant N$ as the explicit overlap will count atoms more times. Hence the complexity of the explicit overlap method has higher complexity than implicit overlap on V2E Module according to the deduction below

$$mn_{exp}^2 \geqslant Nn_{exp} \approx n_{\mathcal{F}}n_{imp}N \tag{40}$$

$$mn_{exp} \approx mn_{\mathcal{F}}n_{imp} \approx n_{\mathcal{F}}N \tag{41}$$

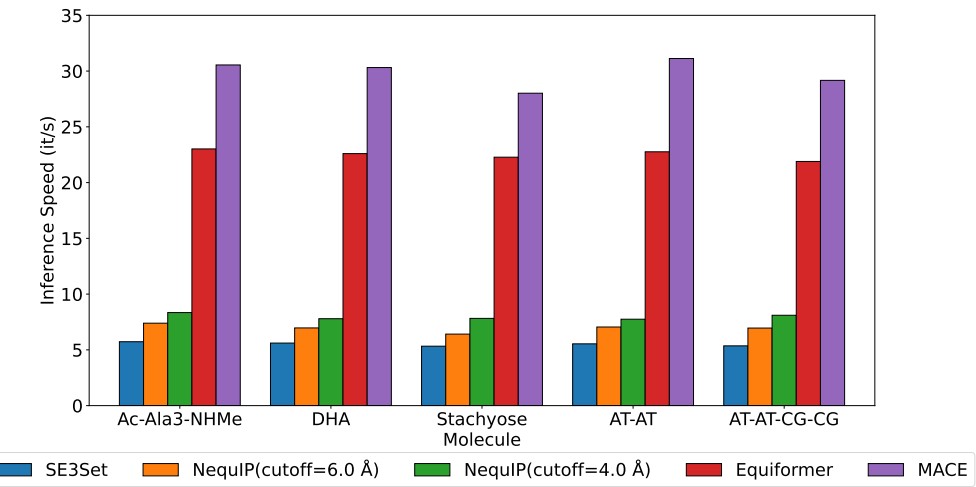

Figure 5: Inference speeds of different models on the MD22 dataset in units iteration/s. The average inference speeds were calculated for each molecule dataset. All tests were performed on a single Tesla A100 80G. The SE3Set includes 3 layers and different cutoff is set (4 Åand 6 Å) for NeuqIP. The other models. The other models use parameter configurations that corresponded to the results reported in their respective studies.

To better support the analysis, we have tested the inference speed of SE3Set on part of QM9 and MD22 datasets as shown in Table 7. Larger fragments, with smaller $c_w$, reduce speed, more so in larger molecules. Speed drops with increased fragment size (smaller $c_w$ or larger $r_c$), and the explicit overlap method is slower than the implicit, leading to our preference for the latter in MD22. This highlights the need to optimize fragment size for a trade-off between detailed interaction capture and speed, particularly in big molecular systems. Additionally, the explicit overlap method is slower than the implicit overlap method, which confirms our previous analysis.

Table 7: The inference speed (unit: iterations/s) of SE3Set with different $c_w$ or $r_c$. The tests on MD22 dataset only use the AT-AT-CG-CG subset.

| | DATASET | EXPLICIT $c_w$ | | | IMPLICIT $r_c$ | | |
|---|---|---|---|---|---|---|---|
| | | 0.1 | 0.05 | 0.01 | 4.0 | 5.0 | 6.0 |
| INFERENCE SPEED | QM9 | 11.60 | 11.54 | 11.37 | - | - | - |
| | MD22 (AT-AT-CG-CG) | 2.62 | 2.29 | 1.68 | 2.92 | 2.90 | 2.87 |

Moreover, we have evaluated the inference time with different baseline models. The results in Fig. 5 show SE3Set runs slightly slower than the other models. This is reasonable as we have included more pairwise node information in our hypergraph. The improvement in our model's performance indicates that our model requires further optimization to achieve a better balance between efficiency and performance.

We also show the number of parameters of SE3Set comparing with some state-of-the-art baseline models as showed in Fig. 8. SE3Set achieves better performance using less parameters.

Table 8: Number of parameters with different models.

| MODEL | VISNET | QUINNET | SE3SET(6 LAYERS) | SE3SET(3 LAYERS) |
|---|---|---|---|---|
| NO. OF PARAMETERS (MILLION) | 10 | 9 | 6.3 | 3.5 |

## K   Training details

This section outlines the training specifics, encompassing the fragmentation parameters, SE3Set hyperparameters, and certain implementation nuances utilized in our experimental setup.

Our dataset construction is founded on PyTorch Geometric (Fey & Lenssen, 2019) augmented with our fragmentation process (Sec. 4.1). Due to inconsistencies in molecular topology identified through RDKit's sanitization routine (Landrum et al., 2020), 1,403 data points were excised from the original dataset. We designated 110,000 data points for the training set and 10,000 for the validation set, selected at random.

We used an explicit overlap scheme on the QM9 and MD17 datasets because of their relatively small molecular systems. We implemented two distinct schemes for calculating fragment bond orders, following either Eq. 18 or Eq. 19. The parameters for fragmentation are detailed below. Given that the MD17 molecules are relatively small, the merge step in the fragmentation process was not actually utilized. However, we present the fragmentation parameters here for the sake of completeness.

Besides, we adopt an implicit overlap scheme on the MD22 dataset and OE62 dataset to reduce computational resource consumption. The details of cutoff $r_c$ introduced in 4* can be found in Table 11.

On QM9 and MD17 dataset, our model was trained using a single Tesla V100 GPU with 32GB of memory, except for the 6-layer model employing the exponential bond order on QM9 dataset, which was trained on two Tesla V100 GPUs with 32GB each. For MD22 dataset and OE62 dataset, our model was trained on a single Tesla A100 GPU with 80GB of memory.

We selected $l = 2$ for our irreducible representations (irreps) feature, which includes both node and hypergraph features. For the radial basis function (RBF), we utilized Gaussian basis functions or Bessel basis functions for the QM9 dataset (Ruddigkeit et al., 2012; Ramakrishnan et al., 2014) and exponential basis functions for the MD17 dataset, MD22 dataset and OE62 dataset (Chmiela et al., 2017). Details could be found in Table 9 and Table 10.

Table 9: Hyper-parameters for training SE3Set model. In the context of hyperparameter settings for dimensions, the symbols $e$ and $o$ are used to denote even and odd parity, respectively.

| Hyper-parameters | QM9 Value or discriptions | MD17 Value or discriptions | MD22 Value or discriptions | OE62 Value or discriptions |
|---|---|---|---|---|
| Optimizer $n_{min}$ | AdamW | AdamW | AdamW | AdamW |
| Learning rate scheduler | Cosine | Cosine | Cosine | Cosine |
| Warm up epochs $n_{max}$ | 5 | 10 | 10 | 10 |
| Minimum learning rate | $1.0 \times 10^{-6}$ | $1.0 \times 10^{-6}$ | $1.0 \times 10^{-6}$ | $1.0 \times 10^{-6}$ |
| Batch size | $32, 128$ | 8 | 8 | 32 |
| Number of epochs | 400 | 1500 | 1500 | 400 |
| Weight decay | $5.0 \times 10^{-3}$ | $1.0 \times 10^{-6}$ | $1.0 \times 10^{-6}$ | $1.0 \times 10^{-6}$ |
| Dropout rate | 0.1 | 0.0 | 0.0 | 0.0 |
| RBF cutoff (Å) | 42.0 | Max distance of used atom pairs | | |
| Basis Type | Gaussian or Bessel | Exponential | Exponential | Exponential |
| Number of Basis | 128 or 8 | 32 | 32 | 32 |
| Number of Blocks | 3 or 6 | 3 or 6 | 6 | 3 |
| Node embedding dimension | $[(128, 0e), (64, 1o), (32, 2e)]$ | | | |
| Hyperedge embedding dimension | $[(128, 0e), (64, 1o), (32, 2e)]$ | | | |
| Attention head dimension | $[(32, 0e), (16, 1o), (8, 2e)]$ | | | |
| Feed forward dimension | $[(384, 0e), (192, 1o), (96, 2e)]$ | | | |
| Output feature dimension | $[(512, 0e)]$ | | | |

## L   Comparison with pre-trained baselines

Pre-trained model has become more popular in molecular representation learning recently. For completeness, we have added the performance comparison with some state-od-the-art pre-trained models in this section. We choose the reported average MAE on 3 targets (HOMO, LUMO, gap) of QM9 as a metric, our model performs better than those pre-trained models on this metic as showed in Table 12.

Table 10: Hyper-parameters for fragmentation. The expand threshold does not work for models training on MD22 dataset because they adapt implicit overlap scheme.

| Bond Order Methods | Bond Order by Lendvay (18) | Frgmentation by Exponential (19) |
|---|---|---|
| minimum atoms number $n_{\min}$ | 2 | 2 |
| maximum atoms number $n_{\max}$ | 6 | 6 |
| isolated threshold $(c_{is})$ | 0.1 | 0.4 |
| expand threshold $(c_w)$ | 0.1 | 0.2, 0.4 |

Table 11: Hyper-parameters for step 4* of implicit overlap scheme in MD22 and OE62 experiments.

| Dataset | MD22 | | | | | OE62 |
| | Ac-Ala3-NHMe | DHA | Stachyose | AT-AT | AT-AT-CG-CG | |
|---|---|---|---|---|---|---|
| distance cutoff $r_c$ (Å) | 5.0 | 4.0 | 4.0 | 6.0 | 6.0 | 5.0 |

Table 12: Performance comparing with some pre-trained models on 3 tasks (HOMO, LUMO, GAP) of QM9 dataset. We use the average mean absolute error performance here following the article of baselines. All the baseline results are extracted from the origin articles (Zhou et al., 2023; Ji et al., 2024; Cao et al., 2023).

| Model | UniMol | UniMol-2 | InstructMol | SE3Set |
|---|---|---|---|---|
| Avg. MAE (meV) | 127 | 95 | 136 | 19 |

## M Comparison with graph neural networks

The hypergraph could incorporate more information than graph with binary edges theoretically, as the hyperedge denotes a set of nodes in graphs, which means graph can be treated as a special case of hypergraph as shown in Fig. 6 (a). To further elaborate on the features and advantages of the hypergraph neural network architecture we use, we show the difference between our model and the graph neural network in this section, using 5 linear arranged nodes as an example in Fig. 6 (b)-(d). The black lines indicate the edges in the origin graph. This structure can correspond to the heavy atoms of the molecule malononitrile ($NC-CH_2-CN$). We focus on the message passing from the orange node to the green node in the graph. For a graph neural network, such message passing requires 4 layers. For the graph neural network with the addition of 3-hop node virtual edges 2 layers are required. For SE3Set, on the other hand, only 2 layers are needed using the explicit overlap method, and only 3 layers are needed for the implicit overlap method, even in the case where the 3-hop nodes are not included inside the hyperedges either. This demonstrates the efficiency of SE3Set in message passing.

## N Summary Statistics

To further evaluate the performance in our different runs on different targets, we defined the summary statistics std. MAE following Gasteiger et al. (2019) which reflects the average error compared to the standard deviation of each target by Eq. 42,

$$\text{std. MAE} = \frac{1}{M} \sum_{m=1}^{M} \left( \frac{1}{N} \sum_{i=1}^{N} \frac{|\hat{y_i}^{(m)} - y_i^{(m)}|}{\sigma_m} \right) \tag{42}$$

with target index $m$ in total $M$ targets, dataset size $N$, the predict value of the model trained by $m$-th target, and the corresponding ground truth $y_i^{(m)}$.

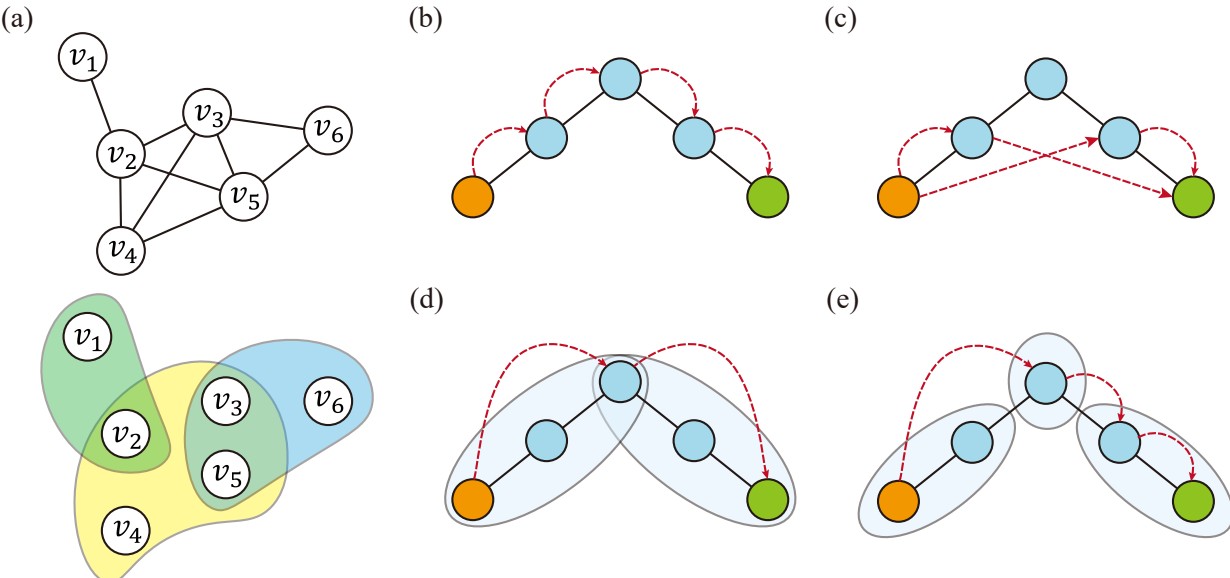

Figure 6: Comparison message passing efficiency of with GNN (graph neural networks), and the GNN adding virtual bodes between 3-hop neighborhoods. The black line between two circles denotes the binay edge in the graph. We consider the message passing process from the orange node to the green node and the nodes in shaded area belongs to a same hyperedge. The red dashed arrows indicate the nodes where the source information reaches after each passing layer. (a) The structural differences between graph (above) and hypergraph (below). The hypergraph contains hyperedges defined by set of nodes. (b) Simple GNN architecture. (c) The GNN architecture adding virtual bodes between 3-hop neighborhoods. (d) The hypergraph in SE3Set builded with explicit overlap method. (e) The hypergraph in SE3Set builded with implicit overlap method.

Furthermore, to evaluate the robustness of the performance especially the MAE, we further propose std. MAE std. by Eq. 43, which reflects the degree of data volatility of MAE on different tasks.

$$\text{std. MAE std.} = \sqrt{\frac{1}{M}\sum_{m=1}^{M}\left(\frac{1}{N}\sum_{i=1}^{N}\frac{|\hat{y}_i^{(m)} - y_i^{(m)}|}{\sigma_m} - \text{std. MAE}\right)} \tag{43}$$

We compute these metrics on QM9 ($M = 12$), MD17 ($M = 7$) and MD22 ($M = 5$) respectively. The OE62 is ignored as there is only one task in its experiment. For some energy related labels ($U_0$, $U$, $H$, $G$) in QM9, the atomic level reference energy is subtracted from the origin value to generate the ground truth. For MD17 and MD22 dataset, as the forces are calculated by the derivation of energy, which means the energy MAE and force MAE belong to the same trained model, we calculate the metrics of energy and force separately with different molecules as different targets. All the results are shown in Table 13. To present the results more clearly, we also show the average value and standard deviation of origin ground truth as well, which presented in the Table 14, 15, 16.

Table 13: The std. MAE and std. MAE std. of different datasets calculated across different targets.

| DATASET | QM9 | MD17 | | MD22 | |
|---|---|---|---|---|---|
| | | ENERGY | FORCE | ENERGY | FORCE |
| STD. MAE (%) | 0.67 | 1.88 | 0.28 | 0.64 | 0.20 |
| STD. MAE STD. (%). | 0.82 | 0.28 | 0.13 | 0.12 | 0.05 |

Table 14: The average groud truth and standard deviation for different targets in QM9 dataset.

| TARGET | $\mu$ | $\alpha$ | HOMO | LUMO | GAP | $R^2$ | ZPVE | $U_0$ | $U$ | $H$ | $G$ | $C_v$ |
|---|---|---|---|---|---|---|---|---|---|---|---|---|
| UNIT | $D$ | $a_0^3$ | eV | eV | eV | $a_0^2$ | eV | eV | eV | eV | eV | $\frac{kcal}{mol\cdot K}$ |
| Avg. | 2.7061 | 75.1917 | -6.5300 | 0.3027 | 6.8328 | 1189.5256 | 4.0415 | -75.9233 | -76.3839 | -76.8206 | -70.6561 | 31.6009 |
| Std. | 1.5304 | 8.1876 | 0.6022 | 1.2772 | 1.2931 | 279.7507 | 0.9054 | 10.3775 | 10.4693 | 10.5440 | 9.5490 | 4.0624 |

Table 15: The average groud truth and standard deviation for different molecules in MD17 dataset. Energy (E) is in unit of kcal/mol and force (F) is in unit of kcal/(mol · Å).

| MOLECULE | ASPIRIN | ETHANOL | MALONALDEHYDE | NAPHTHALENE | SALICYLIC ACID | TOLUENE | URACIL |
|---|---|---|---|---|---|---|---|
| E Avg. | -406737.2765 | -97195.9314 | -167501.8334 | -241898.7881 | -311033.7587 | -170223.8472 | -260107.3102 |
| E Std. | 5.9468 | 4.1920 | 4.1399 | 5.5867 | 5.4163 | 5.0996 | 4.9230 |
| F Avg. | 0.0000 | 0.0002 | 0.0000 | -0.0001 | 0.0000 | 0.0000 | 0.0000 |
| F Std. | 27.9284 | 26.2570 | 28.6374 | 28.6544 | 28.5897 | 27.3297 | 30.0238 |

Table 16: The average groud truth and standard deviation for different molecules in MD22 dataset. Energy (E) is in unit of kcal/mol and force (F) is in unit of kcal/(mol · Å).

| MOLECULE | AC-ALA3-NHME | DHA | STACHYOSE | AT-AT | AT-AT-CG-CG |
|---|---|---|---|---|---|
| E Avg. | -620662.7117 | -631480.1418 | -1578838.9203 | -1154896.6603 | -2329950.4156 |
| E Std. | 8.2038 | 9.5559 | 13.7595 | 10.9584 | 15.7047 |
| F Avg. | 0.0000 | 0.0000 | 0.0000 | 0.0000 | 0.0000 |
| F Std. | 26.0387 | 25.9614 | 25.6038 | 27.9251 | 27.7151 |

