# OpenReview forum: "SE3Set: Harnessing Equivariant Hypergraph Neural Networks for Molecular Representation Learning"
_TMLR — Rejected by TMLR_

### Review · Reviewer_U4i9 · 2024-12-08

**Summary Of Contributions:**

The article presents SE3Set, a hypergraph neural network designed for molecular representation learning, integrating 2D chemical and 3D spatial information through a novel fragmentation method. With SE(3) equivariance and advanced attention mechanisms, SE3Set captures complex many-body interactions and achieves state-of-the-art performance on small molecule datasets, while significantly improving accuracy (~20% reduction in error) for larger molecules in MD22. This approach offers a robust tool for advancing molecular property prediction.

**Audience:**

Yes

**Claims And Evidence:**

No

**Requested Changes:**

- Address the questions and concerns raised in your [ICLR reviews](https://openreview.net/forum?id=dBafcyEQzr).

- For the sake of completeness, include sections that briefly introduce the datasets and baseline models used in the study. Additionally, clarify the source of the baseline performance—whether it comes from their original papers or experiments conducted by the authors. If the results are from prior works, discuss whether the experimental setups are consistent across all baseline methods.

- Provide a more detailed explanation of how hyperparameters were selected. Furthermore, consider discussing the impact of scaling up baseline methods to match the training and inference efficiency of SE3Set. Would such scaling affect their performance, and if so, how?

- Include the MAE variance across different runs to provide a clearer picture of the model's stability and reliability.

- Present micro and/or macro averages for each dataset to provide a more nuanced understanding of performance across diverse scenarios.

- Incorporate the suggested baselines into your study to make the comparison more comprehensive and ensure a robust evaluation of SE3Set's capabilities.

**Strengths And Weaknesses:**

### Strengths

The paper is well-written, and the proposed model, SE3Set, is clearly presented. Integrating hypergraph neural networks with SE(3) equivariance is a solid and innovative idea that holds promise for inspiring future research in molecular representation models.

### Weaknesses

The effectiveness of SE3Set is not convincingly demonstrated. According to the paper, SE3Set underperforms compared to some baselines, such as ViSNet, on datasets featuring smaller molecules like QM9, despite showing improved performance on larger molecules. However, the ablation studies are primarily focused on the QM9 dataset, which seems like a peculiar and questionable choice. Additionally, the authors claim state-of-the-art (SOTA) performance but fail to include comparisons with several of the most advanced and widely recognized molecular representation models, such as Uni-Mol [1] and InstructMol [2]. This omission raises significant concerns about the comprehensiveness and reliability of the study.

Furthermore, I noticed that the same article was submitted to both NeurIPS and ICLR. The ICLR submission is particularly troubling because the authors chose to withdraw it without addressing the reviewers' concerns. Upon checking the feedback provided in the ICLR reviews, I found the critiques reasonable and aligned with my own observations. It is disappointing to see that the manuscript has not been updated to address these issues in its current TMLR submission. This neglect raises concerns about the soundness of the methodology, which is a critical criterion for TMLR submissions according to the [guidelines](https://jmlr.org/tmlr/editorial-policies.html#evaluation:~:text=Are%20the%20claims%20made%20in%20the%20submission%20supported%20by%20accurate%2C%20convincing%20and%20clear%20evidence).

I strongly recommend the authors take their submissions more seriously and dedicate additional effort to improving the quality of their work. Submissions should aim to meet the high standards of rigor expected by the research community, rather than giving the impression of "testing their luck."

Overall, I think the paper needs at least a major revision before being accepted.

[1] Zhou, Gengmo, et al. "Uni-mol: A universal 3D molecular representation learning framework." (2023).
[2] Cao, He, et al. "Instructmol: Multi-modal integration for building a versatile and reliable molecular assistant in drug discovery." arXiv preprint arXiv:2311.16208 (2023).

---

> ### Author Response · Authors · 2024-12-23
>
> We thank the reviewer's criticism.
>
> ### Weakness 1
>
> Thank you for your suggestion. Our ablation experiments include studies on both the QM9 and MD22 molecules, focusing on the impact of the fragmentation method on the SE3Set model and exploring the effectiveness of the SE3Set module design. We propose two fragmentation methods: explicit and implicit. For the QM9 dataset, we used the explicit fragmentation method. However, for the larger molecular systems in the MD22 dataset, we adopted the implicit fragmentation method. We compared different segmentation methods on the QM9 dataset and explored our proposed model architecture. Additionally, we conducted a study on the model's hyperparameters in the MD22 dataset. Since we use explicit overlapping fragmentation on the QM9 dataset, there is a direct difference between this method and the other segmentation methods, and thus it can be compared with existing fragmentation methods such as BRICS. For the MD22 dataset, on the other hand, the fragmentation overlap part of the implicit model is included in the definitions of E2V and V2E modules, and is therefore not suitable for direct comparison with other fragmentation methods.
>
> ### Weakness 2
>
> Thanks for your suggestion. Uni-Mol and InstructMol are pre-trained molecular representation learning models trained on multiple objectives, while SE3Set is designed for single-target prediction. Directly comparing the performance of these models may not be entirely fair. Additionally, it is worth mentioning that InstructMol was accepted by COLING 2025 on November 30, which is after our submission date. At the time of our submission, that paper was still under peer review. Nevertheless, we are willing to compare these baselines with the average MAE for the three tasks (HOMO, LUMO, $\Delta\varepsilon$) reported in [1], [2] and [3]. As shown in the table below, our model outperforms the baseline models.
>
>
> | Model          | Uni-Mol | Uni-Mol 2 | InstructMol-GS | SE3Set(ours) |
> | -------------- | ------- | --------- | -------------- | ------------ |
> | Avg. MAE (meV) | 127     | 95        | 136            | 19           |
>
> ### Weakness 3
>
> Thanks for your suggestion. Regarding your comment about our article being withdrawn from ICLR, according to the TMLR guidelines for reviewers (https://jmlr.org/tmlr/reviewer-guide.html#:~:text=Double%20blind.,from%20the%20submission.),
>
> "Double blind. The TMLR review process is meant to be double blind. The identities of the authors of a submission will be kept from the reviewers, and vice versa. You are expected not to take any actions that would violate this double blind state. This includes not actively seeking to find out the identity of the authors of a submission you are reviewing (e.g., by searching for related presentations online or preprints on arXiv). Generally, we suggest you do searches on related work after doing a preliminary read of the submission. If you accidentally discover the identity of the authors, and believe it may influence your judgment, then contact the Action Editor (AE) so they may determine whether you should be unassigned from the submission.",
>
> your comment is not appropriate. However, we are willing to address the main issues raised by the ICLR reviewers.
>
> 1. Transferability of fragmentation methods.
>
>    In our paper, we have already declared that our fragment-based approach is not applicable to all systems. Nevertheless, it already encompasses a broad application scope for molecules that involve organic/biomolecular interactions with typical covalent bonds. The novelty of our method lies in offering a fragment-based approach that is applicable to the majority of systems for constructing hypergraphs, integrating equivariance with hypergraph neural networks, and attaining substantial enhancements in systems with prominent higher-order interactions, such as in large molecules. This limitation was explicitly addressed in our paper.
>
>    Unlike organic small molecules and biomolecules, the functional groups in the structure of this molecule are not well defined, as the molecule contains a number of aromatic ring structures whose interactions cannot be simply stated in terms of the definitions of functional groups in organic chemistry. Our fragmentation approach currently treats all atoms in the molecule as a hyperedge, which introduces a high level of complexity and is unsuitable for representing SE3Set structures based on hypergraph neural networks.
>
>    For periodic systems, especially those that do not contain explicit forms of covalent bonding, our fragmentation approach is currently unable to deal with them, again due to the definition of the tannic clusters. One possible solution is to define clusters describing the main crystalline electronic properties adding virtual topologies and dividing them within a replicated periodic lattice. We will continue to explore this type of solution in our future work.

---

> ### Author Response · Authors · 2024-12-23
>
> 2. The advantages of our fragmentation methods.
>
>    Our contribution in the fragmentation method section is mainly in two aspects. (1) Introducing 3D conformational information into the fragmentation process to maintain the 3D conformational chemical information of the molecule. (2) Two methods, explicit overlap and implicit overlap, were constructed to ensure that the chemical information between different fragments can be transferred to each other. Both of these features are not available in previous fragmentation methods (e.g., BRICS). Although different fragmentation methods may have their own advantages on other tasks, for hypergraph neural network molecular representation learning in 3D space, as we demonstrate in our ablation experiments, our fragmentation method is more suitable for this scenario.
>
> 3. Give a detailed complexity analysis of SE3Set.
>
>    We have provided a clear analysis in Appendix J of revised manuscript (or in Appendix I in our first submit manuscript), which shows that the complexity of our models both explicit overlap and implicit overlap method grows linearly with system size. Further more, we also give a practice inference speed for SE3Set compared with some baselines.
>
> 4. How to choose the hyper parameters?
>
>    In explicit and implicit overlap, $c_w$ and $r_c$ respectively govern the overlap degree of fragments. The ablation studies on the QM9 dataset indicate that there is no significant difference in predictive error among various $c_w$. Further assessments with varying $r_c$  on the AT-AT-CG-CG dataset show better performance at higher $r_c$ as showed in ablation study. This suggests our network more effectively captures complex interactions with larger fragments, enhancing its predictive capabilities.
>
>    The parameter $n_{\text{min}}$ determines the smallest fragment size. We chose a minimum of 2 to avoid chemically illogical single-atom fragments in our experiments. The choice of parameters is the trade off between efficiency and precision.
>
> 5. Add more experiment to support our opinion.
>
>    To better evaluate the effectiveness of SE3Set, we have supplemented the experiments with the OE62 [4] dataset. The details could be found in Appendix I of our revised manuscript. SE3Set with 3 layers outperforms all the baseline models extracted from [5], which indicate the potential of using SE3Set on large molecules.
>
>    | Model     | SchNet | PaiNN | DimeNet++ | GemNet-T | SE3Set(3 Layers) |
>    | --------- | ------ | ----- | --------- | -------- | ---------------- |
>    | MAE (meV) | 131.3  | 63    | 53.8      | 53.1     | **51.7**         |
>
> 6. The main motivation and novelty of our architecture.
>
>    Many studies have demonstrated that incorporating higher-order many-body interactions is beneficial for molecular representation. However, how to effectively introduce these higher-order interactions remains a challenge and is a topic worthy of further exploration. Considering that the use of hypergraph neural networks can better generalize chemically important substructures, especially functional groups defined in organic chemistry, this improves the model's ability to capture the chemical significance of molecules to a certain extent and benefits the generalization of many-body interactions within and between fragments. Additionally, EGNN has been shown to outperform invariant neural networks in 3D spatial learning tasks, such as molecular conformation representation. However, how to introduce SE(3) equivariance into hypergraph neural networks has not yet been explored. To our knowledge, SE3Set represents the first work combining equivariant and hypergraph neural networks, allowing it to handle higher-order many-body interactions. Our main contribution lies on the three points:
>
>    1. We have developed a completely original method for fragmenting organic molecules. Our fragmentation approach uniquely integrates chemical topology information with 3D spatial data, making it particularly suitable for generating hypergraph representations of molecules. This is an important aspect when developing molecular representations on hypergraphs.
>    2. Since the relative position vector $\vec{r}_{ij}$ is pairwise for nodes, introducing SE(3) equivariance into hypergraph neural networks requires careful design of the interactions between nodes and hyperedges.
>
>    3. We have also extended traditional GNN attention by constructing equivariant attention blocks for feature updates through the tensor product of hyperedges and nodes.

---

> ### Author Response · Authors · 2024-12-23
>
> ### Requested Changes 1
>
> See Weakness 3.
>
> ### Requested Changes 2
>
> The demonstration of our used datasets are included in our experiment section 5.1, section5.2, Appendix H and Appendix I.  The source of the baselines comes from the corresponding original papers. We have added the detailed introduction for the baselines in our revised manuscript in the caption of Table 1, 2, 5.
>
> In summary, SE3Set is trained on 110k QM9 molecules and validation on 10k for QM9 dataset. In MD17 dataset, we randomly select 950 training samples and 50 validation samples. These are consistent with the baseline models. For MD22 dataset, Our dataset partition, consistent with sGDML and QuinNet, designates 500 samples as validation, with the remainder for training.
>
> ### Requested Changes 3
>
> Thanks for your inquiry. The settings for choosing hyper parameters are discussed in Weakness 3-4. We reduced the size of SE3Set by decreasing the size of the SE3Set (6 to 3 layers) in order to achieve the same effect as by increasing the size of the baseline models. Actually, in ablation study of AT-AT-CG-CG in MD22 dataset, we deployed SE3Set using only 3 layers, which corresponds to a reduction in the complexity of our model. Nevertheless, the best results of 3 layers SE3Set can still outperform the baseline models and achieve SOTA performance. This proves that to some extent simply expanding the baseline approach or shrinking our approach by matching does not lead to better results.
>
> ### Requested Changes 4 & 5
>
> To test the MAE variance of different runs to show the the model's stability and reliability, we provide a standard deviation of MAE among different tasks following DimeNet [6]. More over, we test the standard deviation of std. MAE to show the robustness of SE3Set training. The definition and data can be found in Appendix N.  The results prove that our training results are stable for different targets.
>
> We also provide the average and deviation of different datasets in Table 14, 15, 16 of our revised manuscipts. Detailed information refers to our revised manusript Appendix N.
>
> | Dataset            | QM9  | MD17 Energy | MD17 Force | MD22 Energy | MD22 Force |
> | ------------------ | ---- | ----------- | ---------- | ----------- | ---------- |
> | std. MAE (%)       | 0.67 | 1.88        | 0.28       | 0.64        | 0.20       |
> | std. MAE std. (%). | 0.82 | 0.28        | 0.13       | 0.12        | 0.05       |
>
> ### Requested Changes 6
>
> We have added the three baselines discussed in Weakness 2 into our revised manuscript in Appendix L.
>
> [1] Zhou, G., Gao, Z., Ding, Q., Zheng, H., Xu, H., Wei, Z., ... & Ke, G. (2023). Uni-mol: A universal 3d molecular representation learning framework.
>
> [2] Ji, X., Wang, Z., Gao, Z., Zheng, H., Zhang, L., & Ke, G. (2024). Uni-Mol2: Exploring Molecular Pretraining Model at Scale. *arXiv preprint arXiv:2406.14969*.
>
> [3] Cao, H., Liu, Z., Lu, X., Yao, Y., & Li, Y. (2023). Instructmol: Multi-modal integration for building a versatile and reliable molecular assistant in drug discovery. *arXiv preprint arXiv:2311.16208*.
>
> [4] Stuke, A., Kunkel, C., Golze, D., Todorović, M., Margraf, J. T., Reuter, K., ... & Oberhofer, H. (2020). Atomic structures and orbital energies of 61,489 crystal-forming organic molecules. *Scientific data*, *7*(1), 58.
>
> [5] Kosmala, A., Gasteiger, J., Gao, N., & Günnemann, S. (2023, July). Ewald-based long-range message passing for molecular graphs. In *International Conference on Machine Learning* (pp. 17544-17563). PMLR.
>
> [6] Gasteiger, J., Groß, J., & Günnemann, S. Directional message passing for molecular graphs. arXiv 2020. *arXiv preprint arXiv:2003.03123*.

---

### Review · Reviewer_DgtH · 2024-12-09

**Summary Of Contributions:**

This work presented SE3Set, an SE(3) equivariant framework for molecular representation learning. A main theme of this method that distinguishes it from popular GNN based methods is the use of so called hyperedges for modelling atom interactions. To incorporate hyperedges in the model, authors have presented a four-step fragmentation algorithm that seems to break down a molecule into smaller, easier to process subcomponents.

Authors have also compared their methods with other SOTAs, and have obtained promising results.

**Audience:**

Yes

**Claims And Evidence:**

Yes

**Requested Changes:**

Some of the requested changes are already described in the above section. I'd like to see the following additional changes.

1. A simple comparison between HGNN and GNN to clearly illustrate the advantage of hyperedges in molecular property prediction. Complex architecture is not required. I'd like to see the bare-metal comparison, i.e. A comparison between a two layer GNN with virtual edge connecting atoms that are within 3-hop neighborhood and also a two-layer HGNN.  I'd also like to see if a two-layer HGNN does produce better result, how many additional layers (steps of message passing) are needed for GNN to catch up. This will provide a more concrete evidence to support your claim

2. There is no comparison of computational complexity among the baselines, in terms of number of model parameters and prediction speed.

**Strengths And Weaknesses:**

Strengths:
1. This paper is easy to follow and understand
2. Detailed description of the model architecture, implementation and training steps are provided
3. Detailed proof of SE(3) equivariance is provided
4. Achieving good experimental performances.

Weaknesses,
1. There is a lack of easy-to-understand illustration of what a hyper edge is. Is it just a collection of binary edges or something else? I'd appreciate if authors can provide a graphic illustration.
2. One of the main claims is that regular GNN cannot model many body interactions. Authors have used the term "binary interactions" for GNN. This is ambiguous and arguably incorrect. In my understanding, through the use of virtual edges (one extreme case is to make all atoms fully connected via virtual edges) and many layers of graph convolution operation (or message passing), an atom can aggregate information from all other atoms. Is this not many-body interaction? Adding virtual edge in molecular representation has been quite common for achieving high-order interaction among atoms. Please see one example work from Minkai Xu's [An End-to-End Framework for Molecular Conformation Generation via Bilevel Programming].
3. Some important baselines are missing in the comparison. How is the performance of this model compared to Uni-Mol by DeepTech?

---

> ### Author Response · Authors · 2024-12-23
>
> Sincerely thank you for scrutinizing our research. We would address each point in our response respectively.
>
> ### Weakness 1
>
> Thanks for your suggestions. A hyperedge is a collection of some nodes in a hypergraph that generally represents a multivariate relationship between nodes. We have further provided a graphic illustration (Appendix M Fig 6a  in revised manuscript) to demonstrate the difference between hypergraph and graph.
>
> ### Weakness 2
>
> Thanks for your question. You're right. Methods like stacking more message passing layers to convey multi-top information can indeed introduce many-body interactions implicitly. On the other hand, cleverly designed network architectures, such as DimeNet and PaiNN, incorporate angular information to introduce three-body interactions, while GemNet and ViSNet include torsion angles to introduce four-body interactions. Similarly, QuinNet uses dihedral angles to introduce five-body interactions. These methods require careful design to represent many-body interactions explicitly using specific geometric quantities. Designing for higher-order many-body interactions and determining the appropriate geometric quantities to describe them remain challenges. What we want to convey is that a single vanilla message passing neural network layer can only represent two-body interactions. However, by introducing hypergraphs, SE3Set offers a more effective way to explicitly capture higher-order many-body interactions that have chemical significance. We understand your concern, which arises from our lack of clarity. We have differentiated them by using explicit many-body interactions and implicit many-body interactions in the paper.
>
> ### Weakness 3
>
> Thanks for your suggestion. We compared the performance of our model with Uni-Mol and several other pre-trained models by using the average mean absolute error (MAE) of three tasks (HOMO, LUMO, gap($\Delta\varepsilon$)) reported in [1], [2], and [3] on the QM9 dataset. The results are shown in the table below, and it is evident that our results are significantly better than those of the pre-trained models. However, it is important to note that directly comparing a model designed for a specific task with models pre-trained on multiple datasets or multi-task datasets might not be entirely fair.
>
> We have added this section in our revised manuscript in Appendix L.
>
> | Model          | Uni-Mol | Uni-Mol 2 | InstructMol-GS | SE3Set(ours) |
> | -------------- | ------- | --------- | -------------- | ------------ |
> | Avg. MAE (meV) | 127     | 95        | 136            | 19           |
>
> ### Requested Changes 1
>
> Thanks for your suggestion. We can support our claim from a topological perspective.
>
> We use a simple example to demonstrate the difference between HGNN and GNN. Consider a graph consists of 5 nodes and edges  {$(i, i-1)| i > 1$}$\cup${$(i, i+1)| i < 5$} where $i$ is the node index (shown in Appendix M Fig 6). As hypergraph is defined by fragmentation process, here we denote the nodes by heavy atoms in a chemical molecular malononitrile ($\text{NC}$-$\text{CH}_2$-$\text{CN}$) two find the hyperedges which contains less than three nodes so that it may not directly see the virtual edges added in 3-hop neighborhoods.
>
> As shown in Appendix M Fig 6, For the message from orange node to the green node, GNN takes 4 layers to catch the message. Instead, explicit HGNN only takes 2 layer and implicit HGNN takes 3 layers to catch the information. SE3Set is able to achieve the message passing efficiency by adding virtual edges with the same number of layers, even though we didn't add the 3-hop virtual edges directly.  We have added Appendix M in our revised manuscript for more detailed explanation.
>
> ### Requested Changes 2
>
> Thanks for your suggestion. In Appendix J of revised manuscript (or in Appendix I in our first submit manuscript), we provide a detailed complexity analysis, which includes a comparison of computational complexities among the baseline models. We have added the parameter counts for both our model and the baseline models in the manuscript, as shown in the table below. Theoretically, when using a radius cutoff, the baseline models maintain linear complexity, which is the same as SE3Set.
>
> | Model           | SE3Set (3 Layers) | SE3Set (6 Layers) | ViSNet | QuinNet |
> | :-------------- | ----------------- | :---------------- | :----- | :------ |
> | # of Parameters | 3.5M              | 6.3M              | 10M    | 9M      |
>
> [1] Zhou, G., Gao, Z., Ding, Q., Zheng, H., Xu, H., Wei, Z., ... & Ke, G. (2023). Uni-mol: A universal 3d molecular representation learning framework.
>
> [2] Ji, X., Wang, Z., Gao, Z., Zheng, H., Zhang, L., & Ke, G. (2024). Uni-Mol2: Exploring Molecular Pretraining Model at Scale. *arXiv preprint arXiv:2406.14969*.
>
> [3] Cao, H., Liu, Z., Lu, X., Yao, Y., & Li, Y. (2023). Instructmol: Multi-modal integration for building a versatile and reliable molecular assistant in drug discovery. *arXiv preprint arXiv:2311.16208*.

---

> > ### Comment · Reviewer_DgtH · 2024-12-29
> > **Concerns well addressed**
> >
> > I'd like to thank authors for addressing all my requested changes adequately. I have no more questions.

---

> > > ### Author Response · Authors · 2024-12-29
> > >
> > > We are delighted that our response has met with your satisfaction. Your constructive feedback is highly appreciated.

---

### Review · Reviewer_P1Tm · 2024-12-10

**Summary Of Contributions:**

This paper introduces SE3Set, an SE(3)-equivariant hypergraph neural network designed for molecular representation learning. The main contributions include: a hypergraph neural network architecture that captures many-body interactions, and integration of SE(3) equivalence within the hypergraph framework and a novel fragmentation method that combines 2D chemical and 3D spatial information. The method shows similar performance to SOTA on small molecule datasets and claims good improvement on larger molecules in MD22.

**Audience:**

Yes

**Claims And Evidence:**

Yes

**Requested Changes:**

1. On the benchmarks, SE3Set shows notable improvements on MD22 but performs quite similarly to many existing methods on QM9. Since SE3Set is claimed to be more effective on large molecules where many-body interactions are more important. To support this claim well, experiments on more large molecule benchmarks should be provided.

**Strengths And Weaknesses:**

strengths:
1. put hypergraph with SE(3) equivalence is novel and there are existing works support the claim SE(3) could handle molecular symmetries.
2. The ablation studies are comprehensive. showing how model architecture impacts overall performance.
3. Using hypergraph with fragmentation method is very suited for molecules as it captures local function group interaction, and also reduce complexity for long range interaction learning.

weaknesses:
 1. On the benchmarks, SE3Set shows notable improvements on MD22 but performs quite similarly to many existing methods on QM9. Since SE3Set is claimed to be more effective on large molecules where many-body interactions are more important. To support this claim well, experiments on more large molecule benchmarks should be provided.

---

> ### Author Response · Authors · 2024-12-23
>
> Sincerely thank you for scrutinizing our research. To further test the performance of SE3Set at more scales, we supplemented the experiments with the OE62 dataset [1], which contains many energy annotated large organic molecules with 3D conformations. We have added the experiment details and results in the Appendix I, and the results are also shown as below. SE3Set with 3 layers outperforms all the baseline models extracted from [2], which indicate the potential of using SE3Set on large molecules.
>
> | Model     | SchNet | PaiNN | DimeNet++ | GemNet-T | SE3Set(3 Layers) |
> | --------- | ------ | ----- | --------- | -------- | ---------------- |
> | MAE (meV) | 131.3  | 63    | 53.8      | 53.1     | **51.7**         |
>
> [1] Stuke, A., Kunkel, C., Golze, D., Todorović, M., Margraf, J. T., Reuter, K., ... & Oberhofer, H. (2020). Atomic structures and orbital energies of 61,489 crystal-forming organic molecules. *Scientific data*, *7*(1), 58.
>
> [2] Kosmala, A., Gasteiger, J., Gao, N., & Günnemann, S. (2023, July). Ewald-based long-range message passing for molecular graphs. In *International Conference on Machine Learning* (pp. 17544-17563). PMLR.

---

### Review · Reviewer_5Bit · 2025-01-09

**Summary Of Contributions:**

This work first proposed a novel fragmentation method on molecules that considers both chemical and 3D spatial information of the molecular system, and the authors further proposed a novel architecture SE3Set to incorporate equivariance into the hypergraph neural network that works on the fragmented molecule. Empirical results on different data sets demonstrate the effectiveness of proposed method, especially on property prediction on rather large molecules.

**Audience:**

Yes

**Broader Impact Concerns:**

Since molecular property prediction is directly related to many fields like pharmaceutical or chemical industry, I suppose a Broader Impact Statement (which is missing in current submission) should be necessary for any works on this task. The authors may need to thoroughly check their work and see if any possible negative impacts may persist.

**Claims And Evidence:**

Yes

**Requested Changes:**

- **More sufficient arguments on the connection between motivation (modeling many-body interaction) and proposed method (fragmentation and SE3Set).** I am convinced that the novel fragmentation method and SE3Set can better tackle larger molecules, but I am a bit confused on how these methods are related to many-body interaction. Some more explanation may be needed here.

- **Ablation study on other data sets, probably MD22 as the authors have mentioned that it contains many large molecules.** I can only find the ablation study on QM9 data set, which seems a bit confusing as the proposed method does not yield good performance on this data set.

- **More direct comparison on the influence of molecule size.** While the authors tried to claim that the proposed method works better on larger molecules by comparing its performance on different data sets, a more direct comparison should be directly comparing molecules with different sizes inside a single data set, as different data sets may have other differences that can influence the model performance.

- **Some visualization results on the fragmentation method.** While the authors have compared the performance of different fragmentation methods in ablation study, I am also interested in how these methods break down complex molecules. The authors can consider choosing some representative molecules and show how different fragmentation methods lead to different results on them.

**Strengths And Weaknesses:**

Strengths:
- The proposed method is clearly introduced and easy to understand
- The proposed method generally achieves sound performance, especially on data sets with many large molecules

Weaknesses:
- The connection between motivation and proposed method seems not strong enough in current version
- Empirical comparison needs to be more complete and convincing

---

> ### Author Response · Authors · 2025-01-17
>
> Sincerely thanks for your suggestions. We are willing to address each point in our response respectively although the reject decision has been given to this submission.
>
> ### Requested Changes 1
>
> As shown in our paper's introduction, numerous studies indicate that incorporating higher-order many-body interactions leads to more refined molecular representations. However, effectively incorporating these interactions within molecular systems remains a significant challenge. For instance, QuinNet only incorporates up to fifth-order many-body interactions, leaving the incorporation of sixth-order and higher interactions unexplored. While MACE theoretically allows for arbitrary-order many-body interactions, its computational complexity limits practical applications to lower orders.
>
> We posit that a key reason for the difficulty in developing neural networks that incorporate higher-order many-body interactions stems from the prevalent use of graph neural networks (GNNs). GNNs inherently describe explicit pairwise (two-body) interactions, requiring intricate design tricks to capture higher-order relationships. Therefore, we aim to develop a more effective method for incorporating many-body interactions. We find that hypergraphs, as generalizations of graphs, make hypergraph neural networks (HGNNs) a highly suitable tool for this purpose. However, constructing hypergraphs in the context of molecular representation requires careful consideration.
>
> Given that chemically meaningful substructures, such as functional groups defined in organic chemistry, have proven effective in enhancing a model's ability to capture molecular chemical significance and generalize many-body interactions within and between these substructures, and considering that 3D information provides finer details beyond topological structure, we aim to construct hypergraphs by combining 2D and 3D information. This leads us to develop a novel fragmentation method, where these fragmented substructures become hyperedges in the hypergraph.
>
> Furthermore, since our molecular representation task involves 3D structures, and considering that Equivariant Graph Neural Networks (EGNNs) have demonstrated superior performance compared to invariant neural networks in 3D learning tasks such as molecular conformation representation, incorporating equivariance into HGNNs is an unexplored area. To our knowledge, existing works exploring equivariance in HGNNs focus on permutation equivariance, rather than the rotational equivariance required for 3D tasks. We aim to leverage the advantages of EGNNs within our hypergraph framework.
>
> In summary, we introduce a novel fragmentation method for constructing molecular hypergraphs and incorporate equivariance into HGNNs, developing the SE3Set model for molecular hypergraph-based learning tasks. To the best of our knowledge, SE3Set is the first work to combine equivariance with hypergraph neural networks, enabling the effective handling of higher-order many-body interactions.
>
> We have included a more detailed explanation of our motivation in the revised introduction of our manuscript.
>
> ### Requested Changes 2
>
> Thanks your suggestion. In fact, our ablation experiments include studies on both the QM9 and MD22 molecules, focusing on the impact of the fragmentation method on the SE3Set model and exploring the effectiveness of the SE3Set module design. In our work, we have proposed two fragmentation methods: explicit and implicit. For the larger molecular systems in the MD22 dataset, we adapt the implicit fragmentation method. Since we use explicit overlapping fragmentation on the QM9 dataset, which incorporate the overlap method in the definition of the aggregation process in the E2V and V2E modules, it is not appropriate to give a direct comparison to other fragmentation methods like BRICS. Therefore, we give the comparison to other fragmentation methods on QM9 dataset using explicit overlap method.
>
> ### Requested Changes 3
>
> Thanks for your advice. To further evaluate the SE3Set with different molecular size in a same dataset, we have tested the MAEs comparing to the baseline models in different groups seperated by the number of atoms on OE62 dataset [1], as shown in our revised manuscript Sec 5.3. The results showed that SE3Set performs significantly better than the other baseline models on larger molecules. This is reflected in the fact that the MAE growth of SE3Set is significantly lower than the baseline model when the number of atoms increases. Additionally, the relative difference (defined in Eq. 18) indicates the advantage of SE3Set is more apparent on the larger molecular systems. The results further confirm that our model works better on larger molecules.
>
> The details of the results refer to the Sec 5.3 of our revised manuscript.

---

> > ### Author Response · Authors · 2025-01-17
> >
> > ### Requested Changes 4
> >
> > We appreciate your feedback. To illustrate the distinctions and advantages of our fragmentation approach compared to existing methods like BRICS, we present a case study using ethylene glycol diacetate. Our method generates overlapping fragments, facilitating message passing between hyperedges in the resulting hypergraph. Critically, it preserves the integrity of substructures, such as predefined functional groups, within these hyperedges—a property not consistently ensured by BRICS. Furthermore, our method incorporates both 2D and 3D information, ensuring that distance effects are not neglected.
> >
> > Further details of this analysis are provided in Appendix L and Figure 8 of the revised manuscript.
> >
> > ### **Broader Impact Concerns:**
> >
> > Thanks your suggestion. We have included a new section to demonstrate the impact in the revised manuscript.

---

### Decision · Action_Editor_Hg2R · 2025-01-12

**Recommendation:** Reject

**Comment:**

While this submission is potentially interested to many audiences, I make a conservative recomendation to ensure the quality of the revision.

**Audience:**

Yes. AI4Science is a popular topic now. Should be interested to researchers working with molecular representation.

**Claims And Evidence:**

A fragmentation method is used to introduce hyper-edges (which contains both 2D and 3D information) to molecules, a variant of hyper-graph neural network is proposed to guarantee orientationindependent molecular representations.

While the idea is not novel, the method makes sense. Since TMLR does not emphasize on creativity of the idea but on the logic and usefulness of the results, the following concerns are raised by the reviewers, which needs to be further justified.
1. Ablation study on MD22 as the authors have mentioned that it contains many large molecules.
2. Ablation study on on the influence of molecule size.
3. Case study and visulization of fragments.

Besides, I also have below suggestions
1. "fragmentation must maintain functional groups and ring integrity". Why this must hold? There are subgraph GNN methods which can identify important subgraph in the big graphs. These subgraphs can be important for performance but do not necessary to be consistent with functional groups. Specifically, when the functional group is larger the difference can also be larger.
2. Try to include more empirical evidence in the main content. Authors can squeeze Figure 2 / Section 4. Empirical evidence is more emphaszied for TMLR.

**Resubmission Of Major Revision:**

The authors may consider submitting a major revision at a later time.